# The cell envelope of *Staphylococcus aureus* selectively controls the sorting of virulence factors

Xuhui Zheng[1], Gerben Marsman [2], Keenan A. Lacey [1], Jessica R. Chapman [3], Christian Goosmann[2], Beatrix M. Ueberheide[3,4] & Victor J. Torres [1✉]

*Staphylococcus aureus* bi-component pore-forming leukocidins are secreted toxins that directly target and lyse immune cells. Intriguingly, one of the leukocidins, Leukocidin AB (LukAB), is found associated with the bacterial cell envelope in addition to secreted into the extracellular milieu. Here, we report that retention of LukAB on the bacterial cells provides *S. aureus* with a pre-synthesized active toxin that kills immune cells. On the bacteria, LukAB is distributed as discrete foci in two distinct compartments: membrane-proximal and surface-exposed. Through genetic screens, we show that a membrane lipid, lysyl-phosphatidylglycerol (LPG), and lipoteichoic acid (LTA) contribute to LukAB deposition and release. Furthermore, by studying non-covalently surface-bound proteins we discovered that the sorting of additional exoproteins, such as IsaB, Hel, ScaH, and Geh, are also controlled by LPG and LTA. Collectively, our study reveals a multistep secretion system that controls exoprotein storage and protein translocation across the *S. aureus* cell wall.

[1] Department of Microbiology, New York University Grossman School of Medicine, New York, New York, USA. [2] Department of Cellular Microbiology, Max Planck Institute for Infection Biology, Berlin, Germany. [3] The Proteomics Laboratory, Division of Advanced Research Technologies, New York University Grossman School of Medicine, New York, New York, USA. [4] Department of Biochemistry and Molecular Pharmacology, New York University Grossman School of Medicine, New York, USA. ✉email: Victor.Torres@nyulangone.org

In Gram-positive bacteria, the first barrier for a protein to be secreted is a single layer of membrane. The majority of secreted proteins are translocated through this membrane by the Sec translocon[1], and a small fraction of proteins employ specialized mechanisms such as SecA2/SecY2, Tat, or the type VII secretion system (T7SS)[2,3]. The cell wall of Gram-positive bacteria is composed of peptidoglycan, teichoic acids, and a variety of proteins[4]. There is an ongoing discussion on the regulation of protein translocation through the cell wall in Gram-positive bacteria. Although the cell wall is porous, its permeability can be influenced by the chemical and physical modifications of the cell wall components and is dependent on the cargo proteins[5–7].

*Staphylococcus aureus* is a Gram-positive bacterium that can cause a broad spectrum of diseases, ranging in severity from skin and soft tissue infections to life-threatening bloodstream infections[8,9]. The pathogenic lifestyle of *S. aureus* is facilitated by a wide array of virulence factors, including toxins, proteases, adhesins, and immune-modulatory factors[10]. While some of these factors are primarily present in the extracellular milieu, many of them are found at the bacterial cell envelope instead of, or in addition to, being secreted. These include covalently cell wall-bound proteins, which are anchored to the cell wall by the action of sortase[11–13]. Other proteins are bound to the cell envelope non-covalently. While some specific domains that mediate this binding have been characterized[14], the mechanisms by which most proteins non-covalently interact with the cell envelope remain to be defined.

The bicomponent pore-forming leukocidins are crucial components of *S. aureus* virulence as they directly target and kill host cells, including immune cells[15,16]. These toxins are composed of two subunits, designated S- and F-subunit (for "slow" or "fast" based on the chromatography elution profile)[17,18]. Upon binding to specific receptors on host cells, leukocidins form hetero-octameric pores in the plasma membrane of the target cell, resulting in cell lysis[15,16]. While most leukocidins are secreted toxins (e.g. HlgAB, HlgCB, LukED, and PVL), LukAB (also known as LukGH) is located both on the bacterial cell and in the extracellular milieu[19–21]. Using tissue culture models of infection, LukAB is responsible for *S. aureus*-mediated killing of primary human phagocytes[19–23].

Here, we set out to explore how *S. aureus* regulates LukAB secretion as well as the role of bacteria-associated LukAB in host-pathogen interaction. Our data show that bacteria-associated LukAB is an active toxin that can be deployed to target and kill primary human neutrophils. We found that LukAB is located in the cell envelope with a punctate pattern and can be further sorted into two distinct compartments: surface-exposed and membrane-proximal. Genetic screens revealed that the intricate transportation of LukAB across the cell envelope is supported by lysyl-phosphatidylglycerol (LPG) and lipoteichoic acid (LTA). Moreover, we identified additional *S. aureus* secreted proteins, including known virulence factors, that are sorted through a similar pathway as LukAB. Thus, our results highlight the role of *S. aureus* cell envelope in the sorting of selective exoproteins.

## Results

**LukAB is associated with the bacterial cell envelope**. To study the mechanism of LukAB secretion, we used a representative strain of community-acquired methicillin-resistant *S. aureus* USA300, the current epidemic lineage in the United States[24,25]. USA300 produces five bicomponent leukocidins that target human cells[16,26,27]. They are predicted to be translocated across the cytoplasmic membrane through the canonical Sec pathway based on their Sec-type signal sequence at the N-terminus (Supplementary Fig. 1). Previous studies have shown that

leukocidins in the supernatant are processed by the type I signal peptidase at predicted sites, supporting their Sec-dependent secretion[28,29].

We examined the temporal localization of the leukocidins using bacterial cultures grown to the exponential (3 h), early stationary (5 h), and late stationary phases (8 h and 24 h) (Supplementary Fig. 2a). The majority of leukocidins (PVL, HlgAB, HlgCB, and LukED) were primarily detected in the culture supernatant regardless of the growth phase (Fig. 1a and Supplementary Fig. 2b). However, LukA and LukB were both found secreted into the culture supernatant as well as associated with bacterial cells at the exponential phase, whereas at the late stationary phase, LukA and LukB were only detected associated with the bacterial cells (Fig. 1b and Supplementary Fig. 2c). Of note, analysis of a diverse collection of *S. aureus* isolates revealed that the cellular retention of LukAB was conserved across *S. aureus* lineages (Supplementary Fig. 2d).

LukA and LukB are present as heterodimers in the culture supernatant, while the other leukocidins are found as monomers[30]. Therefore, we sought to examine if the LukAB heterodimer complex was responsible for the binding to bacterial cells. To this end, we expressed either *lukA* or *lukB* in a USA300 isogenic mutant strain lacking the *lukAB* operon. Immunoblotting for LukA and LukB demonstrated that each of the subunits alone could be found on the bacterial cell as well as released into the culture supernatant (Fig. 1c). Thus, the unusual localization of LukAB was not due to the formation of a heterodimer toxin complex. As similar cell-association and secretion properties were observed for LukA and LukB, hereafter we focused on the detection of LukA subunit to study LukAB secretion.

We next investigated the subcellular localization of LukAB by fractionating the bacterial cells into cell wall, membrane, and cytoplasmic fractions. To achieve this, the cell wall proteins were liberated by cleaving peptidoglycan with lysostaphin[31]. The resulting protoplasts were then lysed and ultra-centrifuged to separate the membrane and cytoplasmic fractions. In addition to LukA, each fraction was also probed for the cell wall-anchored protein IsdA, the membrane protein sortase A, and the cytoplasmic localized His-tagged GFP as fractionation controls. We noted that a small portion of sortase A was also detected in the cell wall fraction due to inadvertent lysis of protoplasts. Despite this, we were able to conclude that LukAB was enriched in both the cell wall and the membrane fractions, but not in the cytoplasm (Fig. 1d and Supplementary Fig. 2e).

To explore if bacterial-associated LukAB was observed in vivo, a murine intraperitoneal infection model was employed. Indeed, LukAB was found associated with the bacterial cells when USA300 was isolated directly from the peritoneal lavage fluid (Fig. 1e). Notably, a second toxin band of smaller size was also observed, suggesting that LukAB had undergone additional processing in vivo (Fig. 1e).

**Cell envelope-associated LukAB contributes to the cytotoxicity of USA300**. LukAB is critical for *S. aureus* to lyse primary human phagocytes in tissue culture models of infection[19–23]. We next sought to investigate whether the localization of LukAB on the bacterial cells contributes to *S. aureus*-mediated killing of primary human neutrophils (polymorphonuclear leukocytes, PMNs). Firstly, to examine if the cell envelope-associated LukAB was active, USA300 cells were treated with lysostaphin to release the toxin and the solubilized proteins were added to human PMNs. A significant amount of cell death was observed when PMNs were incubated with lysates from USA300 wild-type (WT) but not from the isogenic Δ*lukAB* strain (Fig. 2a). Moreover, the cytotoxicity of WT lysates was neutralized by an anti-LukA antibody[22].

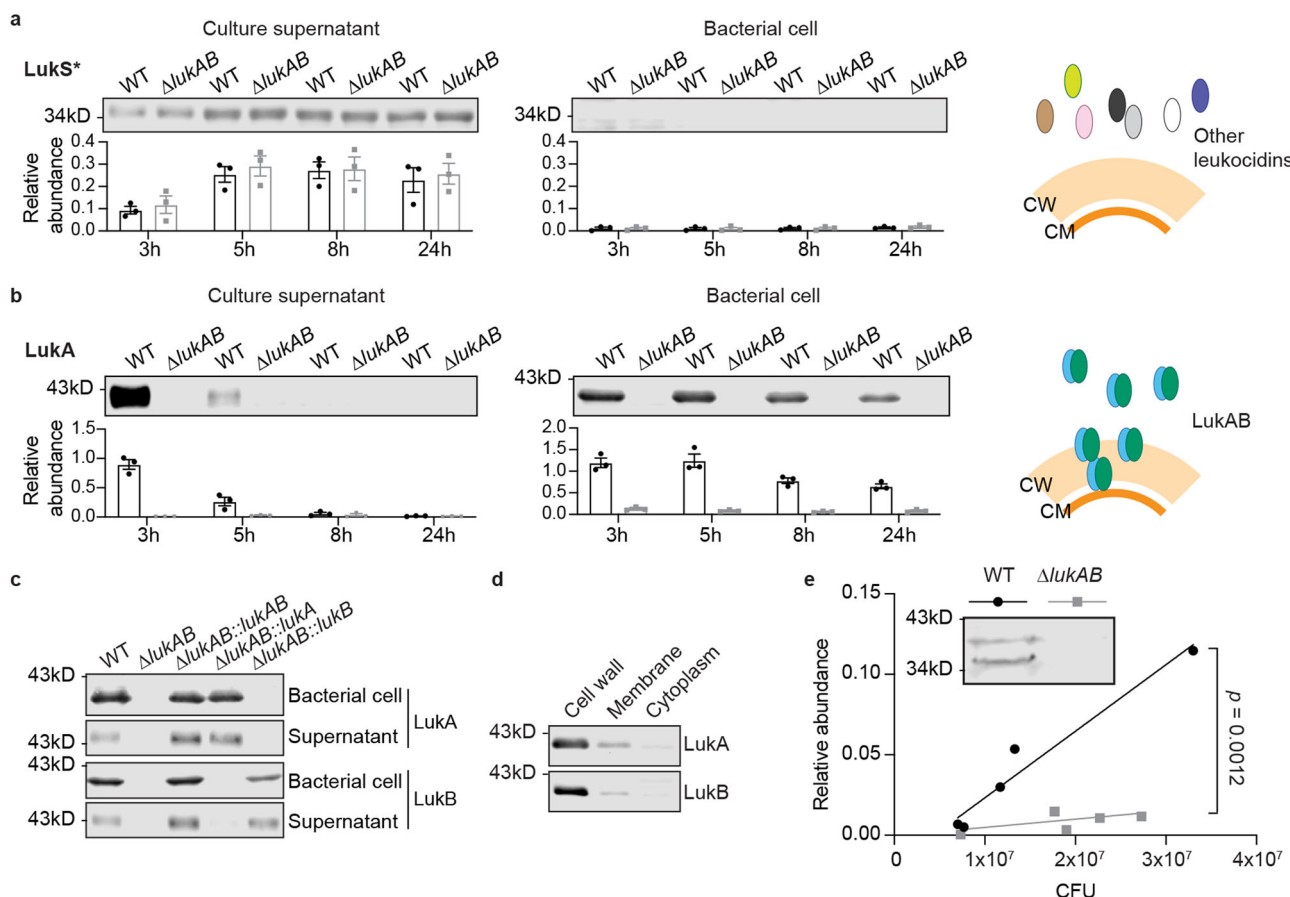

**Fig. 1 LukAB is associated with the bacterial cell. a** Detection of the S-subunit of leukocidins in the culture supernatant and the bacterial cell lysate at different growth phases. Data show a representative immunoblot (top) and mean ± SEM of the protein signal (bottom) from 3 independent experiments. Targeted protein signals were identified and normalized to 50 ng of purified LukE. LukS*, the antibody recognizes LukS-PV, LukE, HlgA, and HlgC. These subunits are highly similar in length and protein sequence. The representative immunoblots were adjusted to the same brightness/contrast. **b** Detection of LukA in the culture supernatant and the bacterial cell lysate at different growth phases. Data show a representative immunoblot (top) and mean ± SEM of the protein signal (bottom) from 3 independent experiments. Targeted protein signals were identified and normalized to 50 ng of purified LukAB. **a**, **b** Cartoon depicts the localization of LukAB and other leukocidins. CM, cell membrane and CW, cell wall. **c** Immunoblot of LukA and LukB in USA300 wild-type (WT) and Δ*lukAB* expressing *lukAB*, *lukA*, or *lukB*. Representative immunoblots of 2 independent experiments are shown. **d** Immunoblot of LukA and LukB in the cell wall, membrane, and cytoplasm fractions of USA300. Representative immunoblots of 3 independent experiments are shown. **e** Detection of LukA in the bacterial cell lysate isolated from mouse peritoneal lavage fluid. The signal intensity of the LukA band was normalized to 50 ng of purified LukAB and plotted against bacterial CFU from each mouse. The *p*-value was determined by an ANCOVA to compare the slopes of the two linear regression lines. The inset shows a representative immunoblot of a pair of WT ($1.3 \times 10^7$ CFU) and Δ*lukAB* ($1.8 \times 10^7$ CFU) samples from the 11 mice infected. Source data are provided as a Source Data file.

Next, we investigated the role of cell envelope-associated LukAB in USA300-PMN interaction using a tissue culture model of infection. To inhibit the synthesis of new LukAB, we employed antibiotics that target protein synthesis. USA300 was pre-treated with either chloramphenicol or tetracycline at >10× the minimum inhibitory concentration, and then co-cultured with PMNs in the presence of the respective antibiotics. The antibiotics inhibited the growth of USA300 both in the absence and presence of PMNs, confirming their activity in this assay (Supplementary Fig. 3a, b). Although infecting PMNs in the presence of antibiotics reduced the cytotoxicity of USA300, antibiotic-treated USA300 killed a significant amount of PMNs in a LukAB-dependent manner (Fig. 2b), suggesting that stored LukAB could be released without active protein synthesis. To further decouple the observed cytotoxicity with the *de novo* production of LukAB, we engineered *lukAB* and *pvl* transcription to be controlled by the *hrtAB* promoter, which is induced by exogenous hemin[32]. Measuring *hrtAB* promoter activity confirmed that protein synthesis was completely inhibited by

tetracycline, even at high concentrations of hemin (Supplementary Fig. 3c). We cultured USA300 with hemin to the early stationary phase and then washed the hemin away. Although the secretion of LukAB and PVL was induced to similar levels (Supplementary Fig. 3d), only LukAB was associated with the bacterial cell (Supplementary Fig. 3e). Pre-synthesis of LukAB but not PVL enabled USA300 to kill PMNs in the presence of tetracycline, further establishing the contribution of cell envelope-associated LukAB in PMN killing (Fig. 2c).

Lastly, we analyzed the impact of growth phase on LukAB-mediated killing of PMNs. Under the examined conditions, we found that the cytotoxicity mediated by cell envelope-associated LukAB decreased from the exponential to the late stationary phase (Fig. 2d).

**LukAB forms discrete foci in the USA300 cell envelope.** To determine the spatial distribution of LukAB on bacterial cells, we immunolabelled LukAB with a polyclonal anti-LukA antibody. Spa and Sbi are two dominant immunoglobulin binding proteins

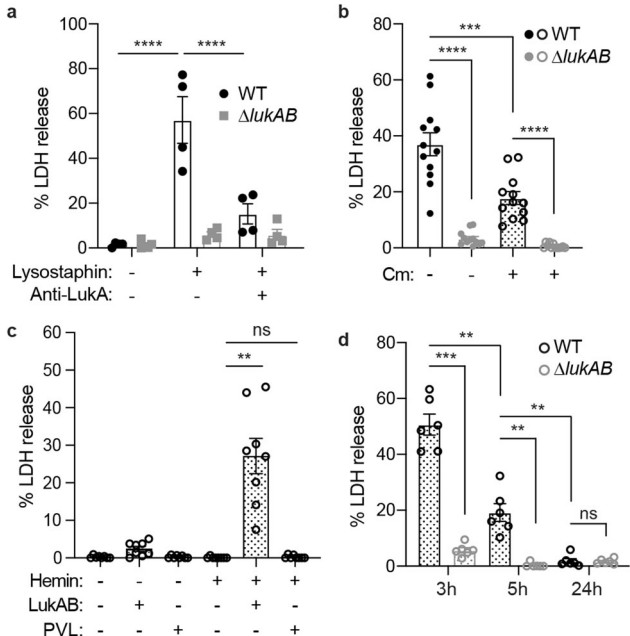

**Fig. 2 The cell envelope-associated LukAB is an active toxin. a** Intoxication of PMNs with cell-free bacterial lysate (Lysostaphin + ) or media control (Lysostaphin -). In the last group, anti-LukA antibodies (Anti-LukA + ) were added into the culture to neutralize LukAB activity. PMN viability was measured by LDH release after 2 h. Bars indicate mean ± SEM of 4 blood donors. ****$p < 0.0001$ by two-way ANOVA with Tukey's multiple comparison test. **b** Infection of PMNs with WT or *ΔlukAB* USA300 ± 100 µg/ml chloramphenicol (Cm) at multiplicity of infection (MOI) = 100. PMN viability was measured by LDH release after 1 h. Bars indicate mean ± SEM of 12 blood donors. ***$p = 0.0005$, ****$p < 0.0001$ by RM one-way ANOVA with Tukey's multiple comparison test. **c** Infection of PMNs with a leukocidin-null isogenic mutant strain complemented by hemin-inducible *lukAB* or *pvl*. The strains were pre-exposed ± 2 µM hemin before the infection. PMNs were infected without hemin, with 40 µg/ml tetracycline at MOI = 25. PMN viability was measured by LDH release after 1 h. Bars indicate mean ± SEM of 8 blood donors. **$p = 0.0057$ by RM one-way ANOVA with Tukey's multiple comparison test. **d** Infection of PMNs with WT or *ΔlukAB* USA300 from the exponential phase (3 h), early stationary phase (5 h), and late stationary phase (24 h) + 100 µg/ml chloramphenicol (Cm) at MOI = 100. PMN viability was measured by LDH release after 1 h. Bars indicate mean ± SEM of 6 blood donors. *p*-values: WT-3h vs *ΔlukAB*-3h, 0.0003; WT-3h vs WT-5h, 0.0014; WT-5h vs *ΔlukAB*-5h, 0.0081; WT-5h vs WT-24h, 0.0076; WT-24h vs *ΔlukAB*-24h, 0.9999, determined by RM one-way ANOVA with Tukey's multiple comparison test. Source data are provided as a Source Data file.

present on the surface of *S. aureus*[33,34], so we used USA300 strains lacking *spa* and *sbi* to minimize unspecific binding of the antibodies. We observed that LukAB was present as discrete foci on the surface of USA300 (Fig. 3a). Over-expressing LukAB from a multi-copy plasmid increased the frequency of LukAB foci but did not change the discrete localization pattern (Fig. 3a). Interestingly, we noticed increased foci frequency as USA300 grew to stationary phase (Supplementary Fig. 4a), which we hypothesized was due to masking of LukAB epitopes by the cell wall. The discrete localization was also observed for FLAG-tagged LukAB by staining with a monoclonal anti-FLAG antibody (Supplementary Fig. 4b). In contrast to LukAB, the staining of a cell wall-anchored protein, protein A, on USA300 lacking *sbi* presented as a continuous signal around the whole cell or hemisphere of the cell (Supplementary Fig. 4c), consistent with the literature[35,36]. To further confirm the

observed distribution with an antibody-free method, we employed the SNAP-tag technology[37]. SNAP-LukA/LukB localized in punctate patterns on the cell envelope of USA300 (Supplementary Fig. 4d, e). Of note, LukAB foci were found both peripherally and near the bacterial septum (Supplementary Fig. 4f). To fully expose the LukAB epitopes that may be masked by the cell wall, we fragmented the cell wall with lysostaphin after the bacteria were fixed, but retained the cell membrane integrity in a high sucrose environment during immunostaining (Supplementary Fig. 4g). Cells that were ruptured during the staining process were excluded from the quantification analysis. Visualization using both confocal microscopy and structured illumination microscopy (SIM) showed a higher frequency of foci after lysostaphin treatment, indicating that lysostaphin exposed LukAB foci on the cell membrane (Fig. 3b and Supplementary Fig. 4h). Furthermore, after exposing the LukAB foci with lysostaphin, the toxin foci were more abundant and of higher intensity at the exponential phase compared to the late stationary phase (Fig. 3c–e).

The localization of cell envelope-associated LukAB was further examined using immunogold labeling with transmission electron microscopy (TEM). LukAB was found in clusters located adjacent to the cell wall and cell membrane (Fig. 3f and Supplementary Fig. 4i). The co-localization of large and small particles, which were probed for LukA and the FLAG-tagged LukA respectively, supported the clustering of LukAB (Fig. 3f). As with the immunofluorescence studies, the LukAB clusters were found in both the septum and the peripheral area (Supplementary Fig. 4i). Altogether, these data demonstrate that LukAB exhibits a dynamic and punctate localization pattern in the cell envelope of USA300.

### LukAB is secreted through a multistep process.

Analysis of the LukAB amino acid sequence using Hmmer[38] and ScanProsite[39] failed to identify any known motifs that mediate cell wall or membrane anchoring. We hypothesized that LukAB was non-covalently bound to the cell envelope and thus examined its solubility by a range of detergents. LukAB could be released by the anionic detergents sodium dodecyl sulfate (SDS) and sarkosyl, but not by nonionic (e.g. Triton X-100, saponin, Tween 20, Brij L23) or zwitterionic (e.g. CHAPS) detergents (Supplementary Fig. 5a). Treating USA300 with SDS did not release the covalently cell wall-anchored protein A, integral membrane protein sortase A, or cytoplasmic protein SaeR (Supplementary Fig. 5b), indicating that USA300 cells remained intact. In addition, LukAB can be solubilized by urea, as well as $MgCl_2$ and LiCl to lesser extents (Supplementary Fig. 5c).

Trypsin is frequently used to remove surface proteins in *S. aureus*[35,40]. Indeed, trypsin treatment cleaved surface-exposed protein A but did not affect membrane-embedded sortase A (Supplementary Fig. 5d). Trypsin removed a portion of LukAB from the bacterial cells, suggesting that this LukAB fraction is exposed on the cell surface (Fig. 4a). To characterize the two fractions of LukAB, we incubated USA300 with trypsin or PBS control, followed by SDS solubilization. After trypsin treatment, no toxin was recovered by SDS, indicating that this fraction of LukAB was susceptible to both trypsin and SDS (Fig. 4b). In contrast, trypsin did not affect the fraction of LukAB that is resistant to SDS solubilization (Fig. 4c). Therefore, we identified two distinct depots of LukAB in the USA300 cell envelope, one exposed on the surface (hereafter referred to as surface-exposed compartment) and the other proximal to the membrane which was protected from trypsin or SDS solubilization (i.e. membrane-proximal compartment) (Fig. 4d).

LukAB was localized predominantly in the membrane-proximal compartment at the exponential phase (3 h), while most LukAB

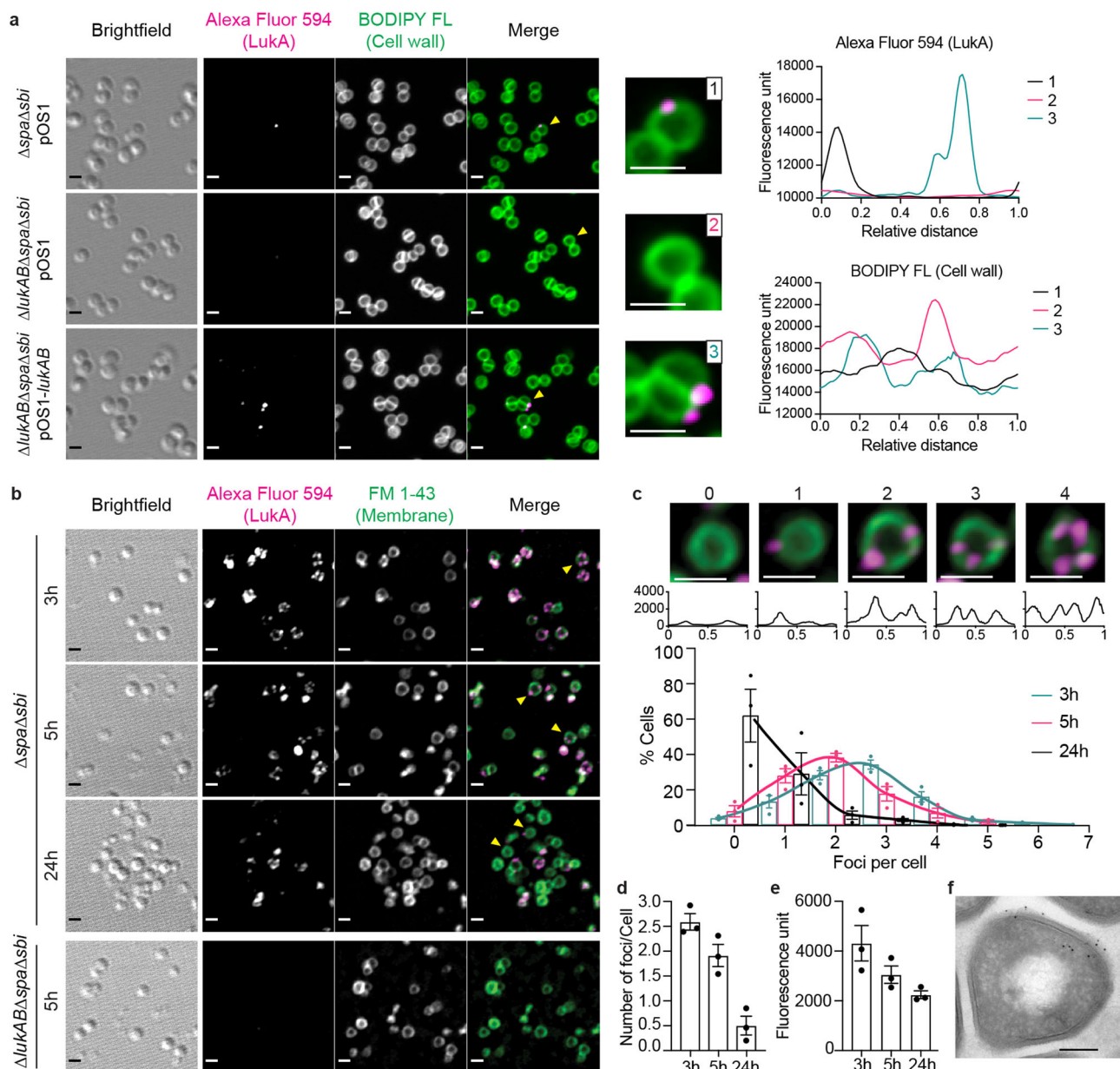

**Fig. 3 Punctate localization of LukAB in the cell envelope of USA300. a** Immunofluorescence imaging of LukAB on USA300 at the early stationary phase (5 h). The pOS1-*lukAB* plasmid over-expressed *lukAB* and pOS1 is the empty vector. The cell wall was stained with BODIPY FL-vancomycin. Yellow arrows point to the single cells shown in enhanced images on the right. For these cells, the fluorescence intensity profiles of the two channels were measured along the cell wall and shown on the right. **b** Immunofluorescence imaging of LukAB on USA300 cells treated with lysostaphin. USA300 grown to the exponential phase (3 h), early stationary phase (5 h), and late stationary phase (24 h) were collected and stained for endogenous LukAB. The cell membrane was stained with FM 1-43. Yellow arrows point to single cells shown in enhanced images in panel **c**. **c** Histogram of the percentage of cells containing different number of foci, using bacterial cultures at the exponential phase (3 h), early stationary phase (5 h), and late stationary phase (24 h). Example images above show cells containing 0–4 LukAB foci. The fluorescence intensity profile of the Alexa Fluor 594 along the cell membrane was plotted below each image. The X-axis is the relative distance from the beginning of the path (top of the cell). The Y-axis is the fluorescence intensity. **d** Quantification of the average number of LukAB foci per cell. **e** Mean fluorescence intensity of each LukAB foci in USA300 Δ*spa*Δ*sbi* strain. **a–c** For fluorescent channels (FM 1-43, BODIPY FL, and Alexa Fluor 594), the maximum projection of Z-stack images is shown. The brightfield image is a single Z slice. Representative images of 3 independent experiments are shown. Scale bar, 1 μm. **c–e** Bars indicate mean ± SEM of 3 independent experiments. Number of cells analyzed combining 3 different fields in each experiment: 3 h, *n* = 231, 290, 523; 5 h, *n* = 161, 195, 129; 24 h, *n* = 92, 319, 136. **f** TEM image showing the immunogold labeling of LukAB. The section was double labeled with antibodies against FLAG-tag (6 nm beads) and LukA (12 nm beads). Scale bar, 200 nm. Representative images of 2 independent EM sessions with different glutaraldehyde concentrations are shown. More TEM images are shown in Supplementary Fig. 4i. Source data are provided as a Source Data file.

was found in the surface-exposed compartment at the late stationary phase (8 h and 24 h) (Fig. 4e). Taken together, these results suggest that during exponential growth, USA300 stores a portion of LukAB inside the cell envelope before the toxin is transported across the cell wall. As bacteria enter the stationary phase, all the LukAB toxin is sorted to the surface of USA300.

Our data suggest a multi-step secretion of LukAB: membrane translocation, sorting within the cell wall, and release into the

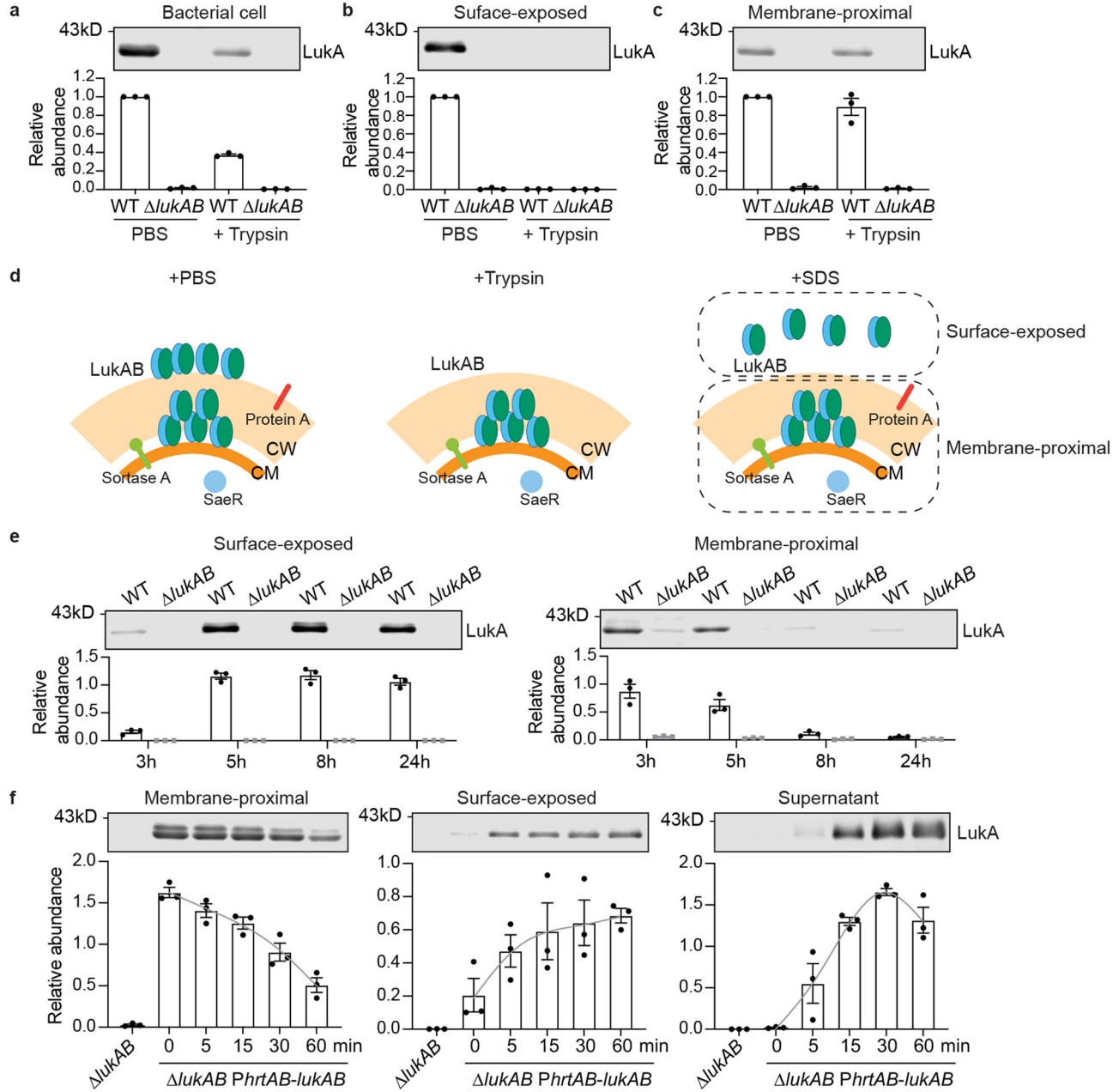

**Fig. 4 LukA is both exposed and unexposed on the surface of USA300. a–c** Relationship between SDS and trypsin treatment. USA300 grown to the early stationary phase was treated with PBS ± trypsin before preparing indicated fractions. Data show a representative immunoblot (top) and mean ± SEM of LukA signal normalized to PBS-treated WT (bottom) of 3 independent experiments. **d** Scheme of the effects of trypsin and SDS. Trypsin removes all surface exposed proteins. SDS solubilizes non-covalently surface-bound proteins, thus the bacterial associated proteins were classified into surface-exposed and membrane-proximal proteins. CM, cell membrane and CW, cell wall. **e** Immunoblot of LukA in the surface-exposed compartment (left) and membrane-proximal compartment (right) in USA300 cells at different growth phases. **f** Immunoblot of LukA in the membrane-proximal compartment, surface-exposed compartment, and culture supernatant. The ΔlukAB strain complemented with hemin-inducible lukAB was pre-exposed to hemin and treated with trypsin before incubated in fresh media. The samples were collected at indicated time. **e**, **f** Data show representative immunoblots (top) and mean ± SEM of protein signal (bottom) of 3 independent experiments. Protein signals were normalized to 50 ng of purified LukA in each experiment. Gray line is the spline fitting of the data. Source data are provided as a Source Data file.

extracellular milieu. To test this model, we accumulated LukAB in the membrane-proximal compartment by inducing LukAB production with exogenous hemin, and then removing surface-exposed LukAB with trypsin. In the time course of 60 min, the LukAB in the membrane-proximal compartment gradually moved to the bacterial surface and the culture supernatant (Fig. 4f). This suggests that the membrane-proximal compartment is an intermediate stage of LukAB secretion.

**MprF and YpfP-LtaA contribute to LukAB sorting.** To identify proteins involved in this multistep secretion process, we screened for mutants that exhibited altered LukAB secretion (Supplementary Fig. 6a and Supplementary Data 1). Using a dot blot assay, we measured surface LukAB levels in 1,920 USA300 mutants from the Nebraska transposon mutant library[41]. Additionally, in a targeted screen, we selected 26 mutants involved in protein secretion, synthesis of cell envelope structures, or other

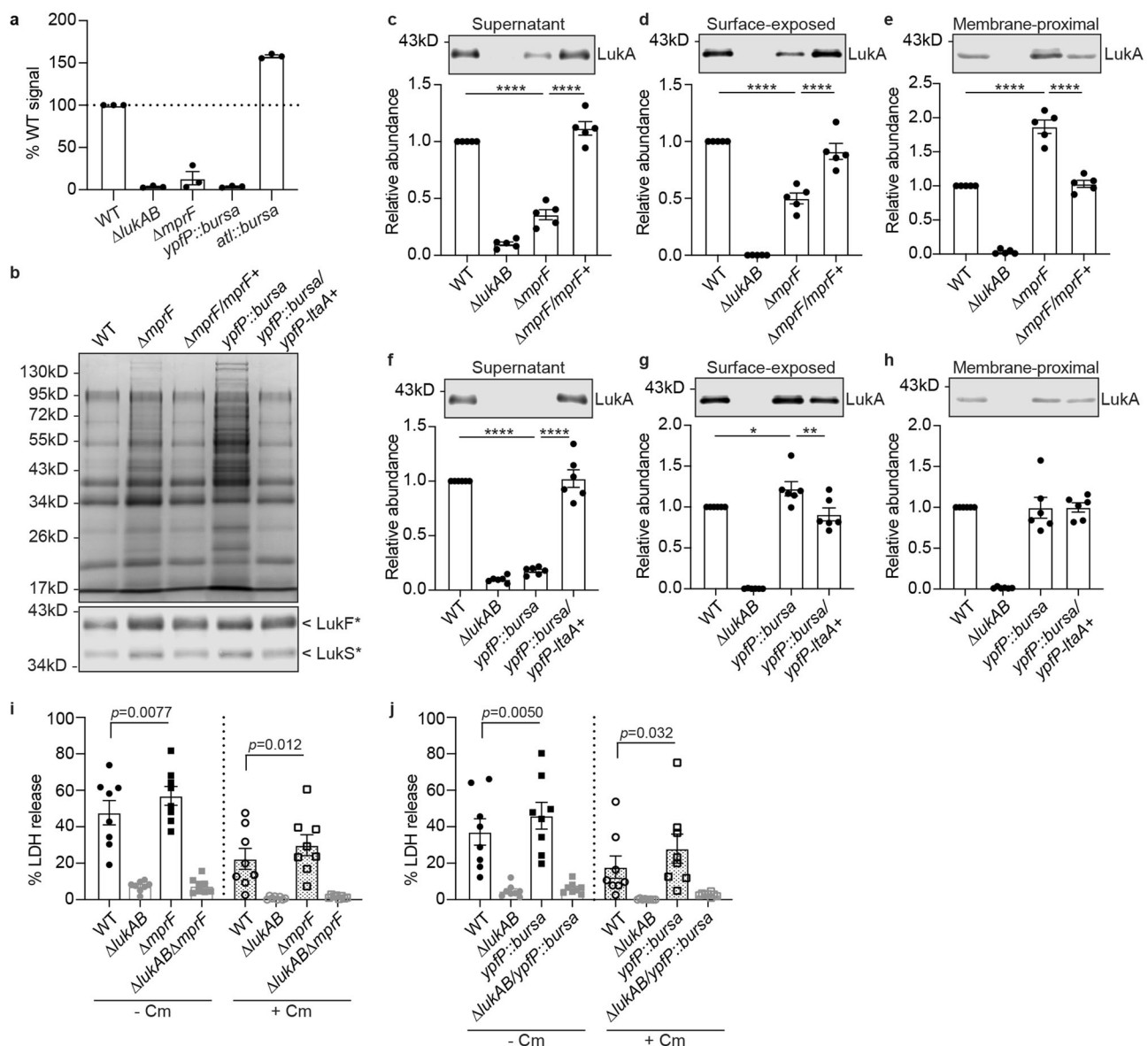

**Fig. 5 MprF and YpfP-LtaA are involved in LukAB deposition. a** Quantification of LukA in the culture supernatant of indicated strains. The LukA signals were quantified from 3 independent immunoblots and normalized to WT. Data show mean ± SEM. **b** Coomassie stained SDS-PAGE of the secreted proteins in the *mprF* mutant, *ypfP* mutant and the respective complement strain (top). Immunoblot of F-subunits (LukF*) and S-subunits (LukS*) of other leukocidins in the culture supernatant in the *mprF* mutant, *ypfP* mutant and the respective complement strain (bottom). LukF*, the antibody recognizes LukF-PV, LukD, and HlgB. LukS*, the antibody recognizes LukS-PV, LukE, HlgA, and HlgC. Representative images of 2 independent experiments are shown. **c–e** Representative immunoblot (top) and quantification (bottom) of LukA in indicated fractions of *ΔmprF* and the complemented strain. Bars indicate mean ± SEM of 5 independent experiments. **f–h** Representative immunoblot (top) and quantification (bottom) of LukA in indicated fractions of *yfpP::bursa* and the complemented strain. Bars indicate mean ± SEM of 6 independent experiments. **c–h** Each dot is an independent experiment. The LukA signal was normalized to WT in each experiment. *$p = 0.032$, **$p = 0.0027$, ****$p < 0.0001$ by RM one-way ANOVA with Tukey's multiple comparison test. **i, j** Infection of PMNs with the *mprF* mutant (**i**) or the *ypfP* mutant (**j**) in the WT or *ΔlukAB* USA300 background. Bacteria and PMNs were co-cultured ± 100 μg/ml chloramphenicol (Cm). Cell viability was measured by LDH release after 1 h. Bars indicate mean ± SEM of 8 blood donors. The *p*-values compared to WT in each treatment were determined by RM one-way ANOVA with Tukey's multiple comparison test. Source data are provided as a Source Data file.

surface modifications, and measured LukAB secretion into the culture supernatant by immunoblotting. From the screens, we identified the *mprF, ypfP,* and *atl* mutants to have aberrant LukAB secretion (Fig. 5a). Herein, we focused on the role of MprF and YpfP in LukAB sorting as both mutants were impaired in LukAB secretion.

In *S. aureus*, MprF mediates the synthesis and translocation of the cationic lipid LPG[42–44]. YpfP (also called UgtP) and LtaA (co-transcribed by the *ypfP-ltaA* operon) synthesize and translocate

diglucosyl-diacylglycerol (Glc$_2$-DAG), which anchors LTA to the cell membrane[45–48]. LPG and LTA are critical components of the *S. aureus* cell envelope. LPG confers positive charge to the cell surface and is important in resistance to many cationic antimicrobial peptides and antibiotics[42,43]. Without YpfP or LtaA, *S. aureus* has abnormally long LTA polymers, which leads to aberrant cell morphology and reduced cell wall integrity[45,49]. Consistent with previous studies, our USA300 *ΔmprF* mutant exhibited decreased positive surface charge compared to WT,

while the ypfP::bursa mutant showed increased positive surface charge, as measured by cytochrome c binding (Supplementary Fig. 7a). We also observed a significant decrease in surface hydrophobicity in the ypfP mutant and a slight reduction in the mprF mutant (Supplementary Fig. 7b). The requirement of MprF and YpfP for LukAB deposition was independent of transcriptional regulation, as lukAB promoter activity was detected at WT levels in the mprF mutant and was elevated in the ypfP mutant (Supplementary Fig. 7c). Notably, while the mprF and ypfP isogenic mutants were impaired in LukAB sorting, the secretion of other exoproteins, including other leukocidins, was increased in both mutants (Fig. 5b), supporting the notion that the general protein secretion machinery was not impaired in the mutants. Importantly, the phenotype observed in the mprF mutant was complemented by introducing a single copy of mprF controlled by its native promoter. Due to the presumably polar effect of the transposon insertion, the phenotype in the ypfP mutant required the full ypfP-ltaA operon to be complemented (Fig. 5b-h).

To investigate how MprF and YpfP-LtaA influence LukAB sorting, we examined the levels of LukAB in the different compartments in the isogenic mprF or ypfP mutant and complement strains. In the mprF mutant, the levels of LukAB in the culture supernatant and surface-exposed LukAB were markedly reduced (Fig. 5c, d). In contrast, increased levels of LukAB in the membrane-proximal compartment were observed (Fig. 5e), suggesting that MprF controls the sorting of LukAB from the membrane-proximal to the cell surface and the extracellular milieu. Enzymatically active MprF was required for this process as the full length MprF with substitutions in the synthase domain (K621A and D731A)[43] cannot complement the phenotype (Supplementary Fig. 7d). Interestingly, the inactive substitutions in the flippase domain of MprF (D71A and E206A)[44] did not affect LukAB release, even though the positive surface charge was impaired in these mutants and the synthase domain alone of MprF failed to complement the LukAB secretion phenotype (Supplementary Fig. 7d).

In the ypfP mutant, LukAB was absent in the culture supernatant (Fig. 5f), but higher levels of LukAB were detected on the bacterial surface (Fig. 5g). The levels of LukAB in the membrane-proximal compartment in the ypfP mutant were more variable and not significantly different compared to WT (Fig. 5h). In addition, GtaB synthesizes UDP-glucose, which is used by YpfP to make Glc$_2$-DAG[47]. We observed that the secretion of LukAB was also impaired in the gtaB mutant (Supplementary Fig. 7e). These results suggest that YpfP-LtaA, via the synthesis of Glc$_2$-DAG, is involved in releasing LukAB from the cell surface to the extracellular milieu.

To explore if MprF and YpfP-LtaA determined the punctate localization of LukAB on the bacteria, we imaged LukAB in the mprF and ypfP mutants. The distribution of LukAB remains punctate around the cell membrane in both WT and the mutants (Supplementary Fig. 7f). Analyses of single-cell images showed an increase in the number of LukAB foci per cell in both mutants (Supplementary Fig. 7g). While the overall LukAB fluorescence intensity is increased in the mprF mutant, the average foci intensity is reduced in the ypfP mutant, suggesting a more dispersed pattern (Supplementary Fig. 7h). Therefore, MprF and YpfP-LtaA do not mediate the punctate distribution of LukAB on the bacterial cells but regulate the distribution pattern.

Since both the mprF and ypfP mutants accumulated LukAB in their cell envelope, we examined their interaction with human PMNs. Both the mprF and ypfP mutants exhibited a slight increase in LukAB-mediated killing of human PMNs compared to WT USA300, in the presence or absence of chloramphenicol (Fig. 5i, j). These results further demonstrate that LukAB stored in the cell envelope contributes to killing PMNs.

**Multistep secretion of other exoproteins**. To determine if additional exoproteins exhibit multistep secretion like LukAB, we performed mass spectrometric analysis on the SDS-released proteome of bacteria grown to the early and late stationary phases (Supplementary Data 2). LukA and LukB were among the most abundant proteins in both growth phases (Fig. 6a and Supplementary Fig. 8a). Most proteins identified were present in all conditions (Supplementary Fig. 8b), while the levels of many proteins were influenced by the bacterial growth phase (Supplementary Fig. 8c). Although highly abundant, the presence of LukAB had a negligible effect on the type or abundance of the other identified proteins (Supplementary Fig. 8b and Supplementary Data 2).

The subcellular localization of the identified proteins was predicted using PSORTb v3.0.2[50]. In addition to LukAB, several predicted extracellular proteins were non-covalently bound to the USA300 cell surface (Fig. 6a and Supplementary Fig. 8d). Most of the highly abundant non-covalently surface-bound proteins were also found secreted into the extracellular milieu (Fig. 6a). Importantly, the protein abundance was not correlated to their relative amount in the culture supernatant (Fig. 6a and Supplementary Fig. 8d), suggesting that the surface association was not a "snapshot" of protein secretion.

To explore if the identified non-covalently surface-bound extracellular proteins follow the multistep secretion described herein for LukAB, we focused on six highly abundant proteins that were predicted to be secreted through the Sec pathway (SignalP-5.0[51]): Geh (SAUSA300_0320), ScaH (SAUSA300_2579), Hel (SAUSA300_0307), SsaA (SAUSA300_2249), IsaA (SAUSA300_2506), and IsaB (SAUSA300_2573). We engineered USA300 strains to produce C-terminal His-tag fusions of these proteins, as well as α-toxin (Hla) as a control, which has low abundance in the surface proteome. In agreement with the proteomics data, all of the selected exoproteins but not α-toxin were found in the surface-exposed compartment associated with the bacterial cell (Supplementary Fig. 9a). In addition, Hel, SsaA, IsaA, and IsaB were also detected in the membrane-proximal compartment, suggesting that these proteins employed a multistep secretion pathway similar to LukAB.

We next characterized the role of LPG and LTA in the sorting and secretion of the selected exoproteins using the mprF and ypfP mutants. Similar to LukAB, IsaB required both MprF and YpfP-LtaA for proper secretion as dictated by the lack of IsaB in the culture supernatant of both the mprF and the ypfP mutants (Fig. 6b). Compared to WT, increased levels of IsaB were found in both the membrane-proximal and surface-exposed compartments in the mprF mutant, and in the membrane-proximal compartment in the ypfP mutant (Fig. 6b). The mprF mutant, but not the ypfP mutant, was deficient in secreting Hel, and more Hel was observed in the membrane-proximal compartment of the mprF mutant compared to WT (Fig. 6c and Supplementary Fig. 9b). In contrast, the release of ScaH and Geh was reduced only in the yfpP mutant (Fig. 6d, e and Supplementary Fig. 9c, d). In the ypfP mutant, increased levels of ScaH were found in the membrane-proximal compartment (Fig. 6d). While the levels of Geh in the membrane-proximal compartment were undetectable, both the pro- and mature-forms of Geh were accumulated on the bacterial surface (surface-exposed compartment) in the ypfP mutant (Fig. 6e). MprF and YpfP-LtaA had no effect on sorting SsaA and IsaA despite more SsaA was produced in the ypfP mutant (Supplementary Fig. 9e, f). Collectively, these data demonstrate that MprF and YpfP-LtaA, likely in the form of LPG and LTA, control the sorting of LukAB, IsaB, Hel, ScaH, and Geh (Fig. 7).

## Discussion

In this study, we investigated the non-conventional secretion of the S. aureus toxin LukAB. We show that LukAB exhibits a

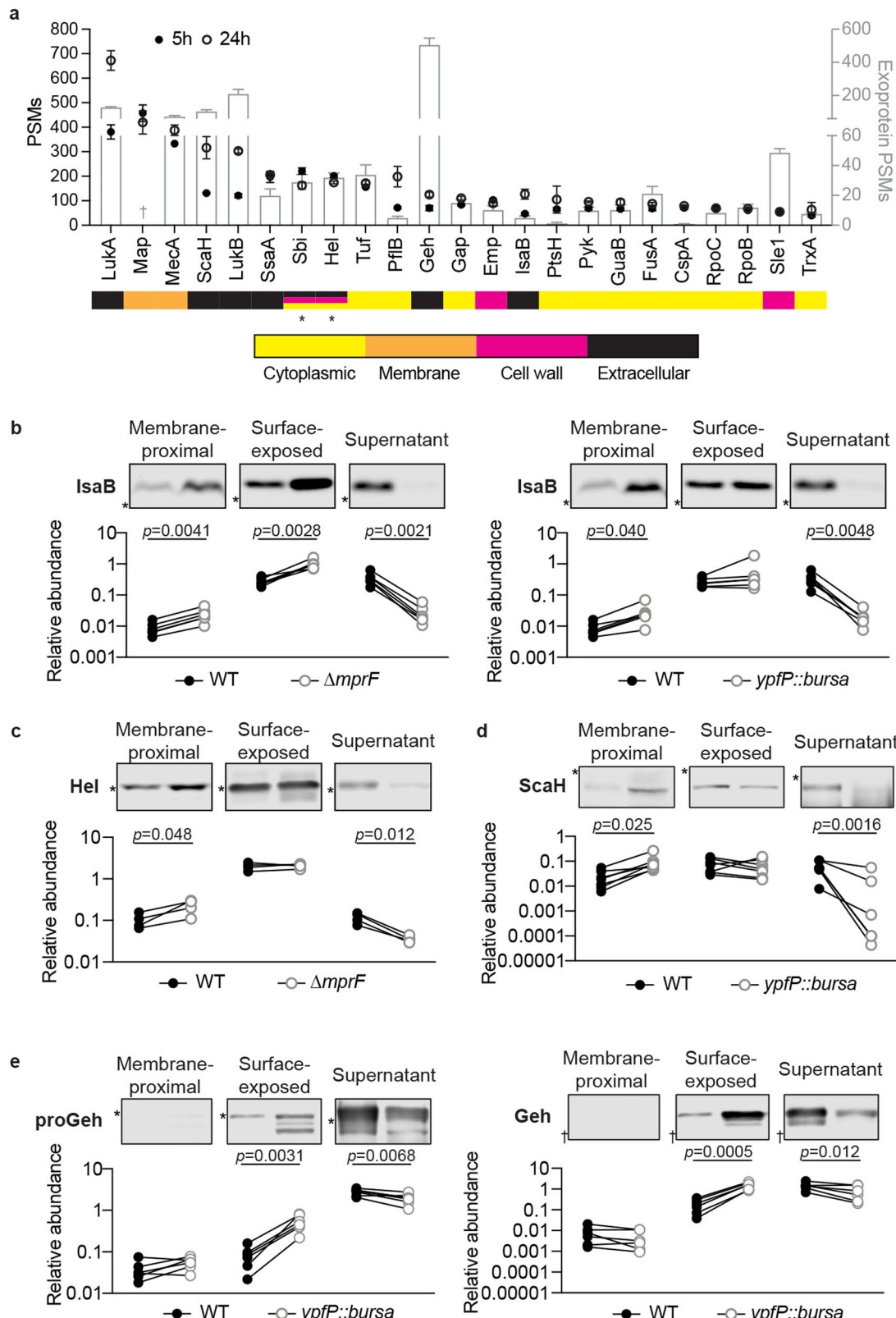

growth phase-dependent distribution in the cell envelope and the culture supernatant. The importance of bacteria-associated LukAB was established by infections with primary human PMNs. Through whole-genome and targeted screens, we identified the cell envelope structures LPG and LTA as required for proper deposition of LukAB. Importantly, the multistep secretion described here is not specific for LukAB as we identified a group

of exoproteins that employed a similar sorting mechanism. Taken together, our data highlight a protein depot in the cell envelope that stores functional virulence factors, as well as the contribution of cell envelope components in sorting exoproteins across the cell wall en route for their release into the extracellular milieu.

Unlike other *S. aureus* secreted toxins, LukAB is found associated with the bacterial membrane and cell wall. The distribution

**Fig. 6 The secretion of non-covalently surface-bound proteins in USA300. a** Most abundant non-covalent proteins on the surface of USA300 identified by mass spectrometry. Proteins with average peptide spectrum matches (PSMs) > 50 are shown here. Circles represent the PSMs of proteins on the USA300 surface (left axis). The PSMs of these proteins in the culture supernatant identified in a previous study[27] are shown in bars (right axis). Data show mean ± SEM of 3 independent samples. The color bars below indicate the localization of each protein predicted by PSORTb. * Sbi and Hel have equal prediction scores localized to the membrane, cell wall, or extracellular. † Data for secreted Map is not available. In Chapman et al., data were searched against a USA300_FPR3757 proteome database where Map protein is not present. **b** Representative immunoblot (top) and quantification (bottom) of His-tagged IsaB in indicated fractions in WT and the *mprF* mutant (left) or the *yfpP* mutant (right) from 6 independent experiments. *, protein marker of 17 kD. **c** Representative immunoblot (top) and quantification (bottom) of His-tagged Hel in indicated fractions in WT and the *mprF* mutant from 4 independent experiments. *, protein marker of 34 kD. **d** Representative immunoblot (top) and quantification (bottom) of His-tagged ScaH in indicated fractions in WT and the *yfpP* mutant from 7 independent experiments. *, protein marker of 95 kD. **e** Representative immunoblot (top) and quantification (bottom) of His-tagged proGeh and Geh in indicated fractions in WT and the *yfpP* mutant from 6 independent experiments. *, protein marker of 95 kD. †, protein marker of 43 kD. **b–e** Each dot is an independent experiment. The protein signals were normalized against 50 ng purified His-tagged LukAB in each experiment. The *p*-values were determined by two-tailed paired *t* tests. Source data are provided as a Source Data file.

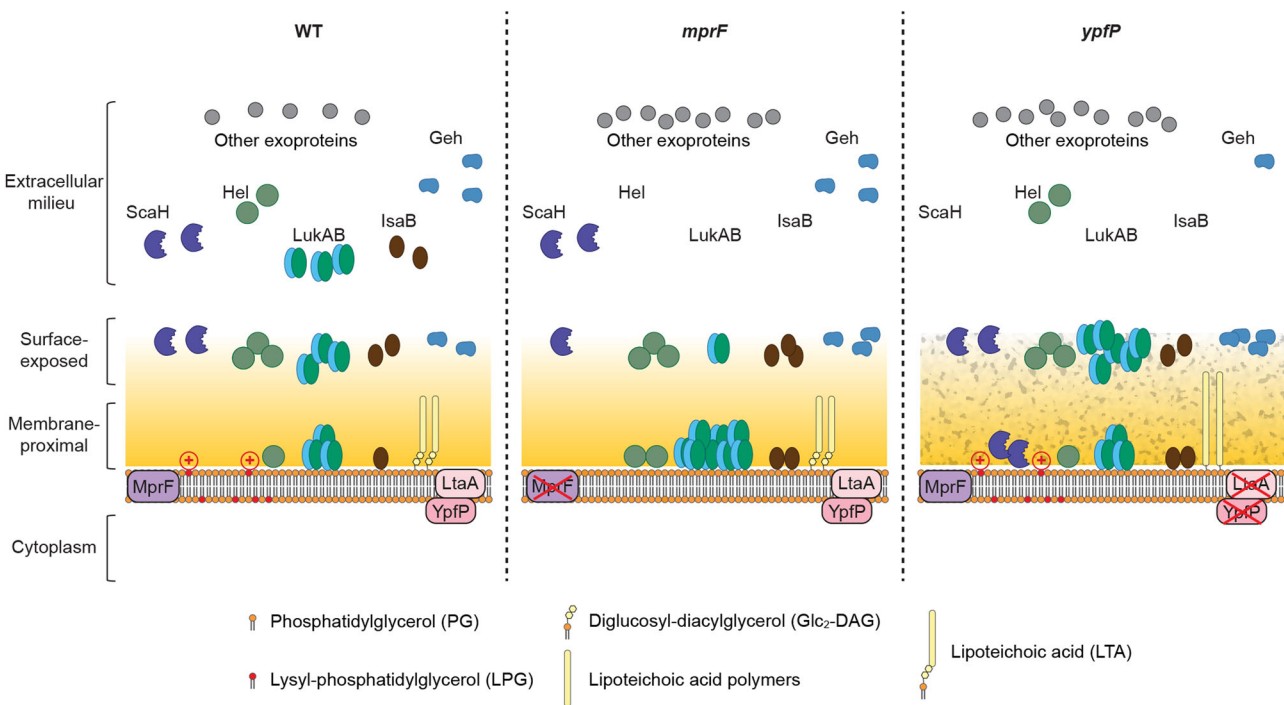

**Fig. 7 Model of MprF and YpfP-LtaA activity in protein secretion.** MprF synthesizes LPG on the cell membrane, conferring positive charge on the bacterial surface. YpfP-LtaA provides the membrane anchor for LTA, and are important in maintaining surface hydrophobicity (indicated by the roughened cell wall in the *ypfP* mutant). In the *mprF* and *ypfP* mutants, while the general protein secretion is functional, the secretion of selected proteins is hampered. Specifically, LukAB, Hel, and IsaB are stuck inside the cell envelope (the membrane-proximal compartment) in the *mprF* mutant. In the *ypfP* mutant, LukAB and Geh are accumulated on the cell surface, while ScaH is stored in the membrane-proximal compartment.

of LukAB on bacterial cells exhibits a punctate pattern. Previously, several proteins have been characterized to be discretely localized in the membrane of *S. aureus*, including the functional membrane microdomain FloA and its cargo proteins, as well as the lipid synthesis proteins PlsY, PgsA, and Cls2[52–56]. Of note, LukAB does not interact with the FloA microdomain[54]. While different localization patterns of the Sec apparatus have been reported in Gram-positive bacteria[57–60], *S. aureus* SecY and SecA appear to be localized evenly on the cell membrane[36,55]. Our data indicate that the punctate distribution of LukAB is regulated by LPG and LTA. Future studies are necessary to explore the components involved in LukAB clustering as well as the LukAB motif(s) responsible for its binding to the membrane and the cell wall.

Membrane microdomains as well as other cell surface components can modulate protein secretion. For example, the flotillin microdomain influences the T7SS, by facilitating the assembly of the machinery[52]. Alpha-type phenol-soluble modulins (PSMα)

can induce the release of membrane lipoproteins and cytoplasmic proteins through increased extracellular vesicle formation as well as cell lysis[61–63]. Wall teichoic acid modulates the spatial distribution of the major autolysin Atl[64,65]. In addition, autolysis is important in releasing multiple cytoplasmic proteins[66]. LukAB is translocated through the cell membrane by the Sec pathway but remains associated with the bacteria in two distinct compartments in the cell envelope before being released. We ruled out PSMs and flotillin as contributors to LukAB secretion and cell association in our targeted screen. Instead, we identified MprF and YpfP-LtaA, which synthesizes LPG and anchors LTA, respectively, as important factors for LukAB transport from the membrane-proximal compartment to the bacterial surface and to the extracellular milieu. Particularly, we observed reduced positive surface charge in the *mprF* mutant and increased positive surface charge in the *ypfP* mutant, as well as a significant reduction in surface hydrophobicity in the *ypfP* mutant (Supplementary Fig. 7a, b). These changes in surface charge and

hydrophobicity could be involved in sorting LukAB through electrostatic and hydrophobic interactions. Another possibility is that LPG and LTA interfere with LukAB binding to an essential component of the cell envelope, and therefore the lack of LPG or LTA results in the retention of more LukAB with the bacteria in the mutant cells. Of note, we only examined mutants present in the arrayed Nebraska transposon mutant library[41]. Other cell envelope structures and/or proteins not present in the library could also be crucial for the secretion of LukAB.

The multistep and cell envelope-controlled secretion described here is not a universal mechanism for all *S. aureus* exoproteins nor specific to LukAB. Through proteomics, we identified additional exoproteins that are non-covalently associated with the bacterial cell envelope, some of which are found in both the surface-exposed and the membrane-proximal compartments. Many of the most abundant proteins identified here, including proteins predicted to be cytoplasmic, have been observed to be associated with the cell envelope in other *S. aureus* strains[67,68], indicating that our findings are not USA300-specific. By examining other exoproteins in the *mprF* and *ypfP* isogenic mutant strains, we found that IsaB, Hel, ScaH, and Geh are secreted via a multistep process similar to LukAB. Interestingly, these proteins are diverse in net charges and functions. IsaB can bind to extracellular DNA and inhibit autophagosome maturation[69,70], while Hel is a putative 5'-nucleotidase. ScaH (also named Aly) is a glycosaminidase and is the only one with a negative net charge at neutral pH among these proteins. The *scaH* mutant exhibits a defect in cell separation when two other cell wall hydrolases, Atl and SagA, are absent[71,72]. Geh (also named SAL2) is a lipase and both proGeh and mature Geh are functionally active[73]. Geh allows *S. aureus* to utilize host-derived lipids as well as to blunt TLR2 signaling by inactivating microbe-derived lipoprotein ligands[73–75]. Intriguingly, the signal peptide of Geh contains a YSIRK/GXXS motif which likely directs this protein to the septum for membrane translocation[76], and recently LTA synthesis has been implicated in this process[36,77]. Of note, we noticed that the effects of MprF and YpfP-LtaA on the sorting of LukAB, IsaB, Hel, ScaH, and Geh can be different, indicating that protein-specific features also play a role during trafficking through the cell envelope.

Altogether, the diverse functions of the proteins characterized in this study suggest that the multistep secretion pathway described here contributes to different aspects of *S. aureus* physiology and pathogenesis. Given that charged phospholipids (i.e. LPG) and LTA are conserved structures of the cell envelope in Gram-positive bacteria, the secretion strategy described here might be exploited by other bacteria. We envision that this system creates a depot in the cell envelope for critical proteins, including virulence factors, to be masked from host immune surveillance, while available to be fired rapidly in specific environments.

## Methods

**Bacterial growth conditions**. *S. aureus* strains were grown on tryptic soy agar (TSA) or tryptic soy broth (TSB). When appropriate, chloramphenicol was added for plasmid selection at 10 μg/ml in the overnight culture or 5 μg/ml in the subculture, and 2 μM hemin was used to induce P*hrtAB* promoter activity. *E. coli* DH5α or IM08B[78] was used for cloning and was grown in Luria-Bertani broth with appropriate antibiotics. *S. aureus* cultures were grown in 5 ml of medium in 15 ml tubes shaking with a 45° angle at 37 °C, except that for mass spectrometry samples, 20 ml of medium in 50 ml tubes were used. For all experiments, *S. aureus* was grown overnight in TSB and a 1:100 dilution of overnight cultures was subcultured into fresh TSB. Unless otherwise specified, *S. aureus* grown to early stationary phase was collected and normalized to OD600 of ~1.1 for further experimental analysis.

**Construction of bacterial strains and plasmids**. *S. aureus* strains, plasmids, and primers used in this study are listed in Supplementary Data 3. *S. aureus* strain

USA300 LAC clone AH1263[79] was used as wild-type (WT) unless otherwise specified.

The LAC *srtA::bursa*, *spa::bursa*, *ypfP::bursa*, *atl::bursa*, *gtaB::bursa* and *esaA::bursa* mutant strains were generated by transducing the mutations of respective JE2 mutant strains from the Nebraska transposon mutant library[41] into LAC with phage 80α. The *sbi::kan* mutant was generated by exchanging the *bursa aurealis* transposon insertion of JE2 *sbi::bursa* with a kanamycin resistant marker using pKAN[80]. The *sbi::kan* was subsequently transduced into LAC *spa::bursa* or Δ*lukAB/spa::bursa*. The *bursa aurealis* transposon insertion of *ypfP::bursa* was exchanged with a spectinomycin resistance marker using pSPC[80] and the resulting *ypfP::spec* was transduced into LAC Δ*spa*Δ*sbi* to generate Δ*spa*Δ*sbi/ypfP::spec*. The Δ*spa*Δ*sbi/ypfP::bursa* strain has the same LukAB secretion phenotype as *ypfP::bursa*.

The Δ*mprF::kan* mutant was generated by replacing the *mprF* locus in LAC WT with *aphA3* gene encoding kanamycin resistance using the pIMAY allelic exchange system[81]. The Δ*mprF::kan* mutation was transduced into isogenic mutants with phage 80α to generate Δ*lukAB*Δ*mprF* and Δ*spa*Δ*sbi*Δ*mprF*.

Chromosomal expression of *lukA*, *lukB*, *lukAB*, *mprF* (including the flippase or synthase domain alone), *ypfP-ltaA*, *geh-6xhis*, *scaH-6xhis*, *hel-6xhis*, *ssaA-6xhis*, *isaA-6xhis*, *isaB-6xhis* and *hla-6xhis* were achieved using plasmid pJC1111, which is stably integrated into the SaPI1 site[82]. In short, PCR amplicons containing the gene and the native promoter were generated and cloned into pJC1111. The resultant plasmid was integrated in strain RN9011 and subsequently transduced into LAC strains with phage 80α. For *mprF* point mutants, selected amino acids were replaced by site-directed mutagenesis using the QuikChange kit (Agilent) using pJC1111-P*mprF-mprF* as the template. As controls, the pJC1111 empty vector was transduced into LAC WT, Δ*lukAB*, Δ*mprF::kan*, and *ypfP::bursa*.

Complementation with FLAG-tagged LukAB was generated by inserting a 3x FLAG tag into the N-terminus of mature LukA on a multicopy pOS1 plasmid[83]. The resultant plasmid was transformed into LAC Δ*lukAB*Δ*spa*Δ*sbi*.

Hemin-inducible *lukAB* and *pvl* were generated by ligating the PCR amplicon containing *lukAB* or *pvl* into the pOS1-P*hrtAB* plasmid[32]. The empty vector and resultant plasmid were transformed into LAC Δ*lukAB* or Δ*leukocidins*. The SNAP-tagged LukAB was generated by fusing a gBlock containing a codon-optimized *snap*-tag from pSNAP-tag(T7)-2 (NEB) with a SGSGSGGRASGSGSG linker to the N-terminus of mature LukA on the pOS1-P*hrtAB* plasmid[32].

**Growth curve**. To measure the growth curve of *S. aureus*, overnight cultures were sub-cultured 1:100 in 150 μl of TSB. The diluted cultures were grown for 24 h at 37 °C using the Bioscreen C (Oy Growth Curves Ab Ltd, Finland). The OD600 was measured every 30 min for 24 h.

**Fractionation of bacterial culture**. When the bacteria were grown to the indicated growth phase, cultures were normalized by OD600 and pelleted by centrifugation at 4,000 rpm (~3,200 × g) for 10 min. The culture supernatant was filtered and precipitated at 4 °C overnight using 10% (v/v) trichloroacetic acid (TCA). The precipitated proteins were washed with 100% ethanol, pelleted, air-dried, and solubilized with 8 M urea for 30 min at room temperature. The solution was mixed with 2x SDS sample buffer, boiled for 10 min, and stored at −80 °C. To prepare the bacterial cell lysate, 1 ml of the normalized bacterial culture was washed with PBS and lysed with 100 μg/ml lysostaphin (Ambi Products LLC), 40 u/ml DNase, 40 μg/ml RNase A, 1x Halt Protease Inhibitor (ThermoFisher) in Lysis buffer (10 mM MgCl₂, 1 mM CaCl₂, 50 mM Tris, pH 7.5) for 30 min at 37 °C. The lysate was mixed with 4x SDS sample buffer, boiled for 10 min, and stored at −80 °C.

To separate surface-exposed and membrane-proximal compartments, the washed bacterial pellet was incubated with 1x Protease Inhibitor in PBS for 10 min at 37 °C, followed by incubation with 1% (w/v) SDS, 1x Protease Inhibitor in PBS for 30 min at room temperature. After centrifugation at 13,000 rpm (~16,000 × g) for 2 min, the supernatant was collected as the surface-exposed fraction. The resulting pellet, namely the membrane-proximal fraction, was washed three times with PBS containing 1 × Protease Inhibitor and lysed as described above. Both fractions were mixed with 4× SDS sample buffer, boiled for 10 min, and stored at −80 °C. The effect of different detergents was tested in the same way as obtaining the surface-exposed fraction, except 1% (v/v or w/v) of Sarkosyl, Triton X-100, Saponin, Tween 20, Brij L23, or CHAPS was used. The effect of 8 M Urea, 50 mM MgCl₂, or 1.5 M LiCl was tested by incubating LAC with indicated reagents in 25 mM HEPES, and followed by the same protocol as obtaining the surface-exposed fraction.

When trypsin was used to remove surface proteins, the washed bacterial pellet was incubated with 0.5 mg/ml trypsin in PBS for 30 min at 37 °C. The untreated sample was incubated with 1× Protease Inhibitor in PBS. Trypsin was neutralized by adding Protease Inhibitor to 1× and incubated for another 5 min at 37 °C. Trypsin was removed by washing the bacterial pellet three times with PBS and the bacterial pellet was lysed as described above.

To examine the translocation of LukAB from the membrane-proximal compartments to the cell surface and supernatant, LAC Δ*lukAB* complemented with hemin-inducible *lukAB* was grown in the presence of 10 μM hemin for 5 h. The bacterial pellet was washed, treated with trypsin for 30 min at 37 °C, and washed 3x with 1x Protease Inhibitor in PBS. The bacterial samples were collected 0, 5, 15, 30, and 60 min after incubating in TSB at 37 °C. Colony forming units

(CFU) were enumerated at the beginning and the end of the experiment, and no significant bacterial growth was observed.

To isolate cell wall, membrane, and cytoplasmic proteins, the bacteria were washed, normalized to $1 \times 10^9$ CFU/ml, and incubated in 0.2 μg/ml lysostaphin, 1× Protease Inhibitor in TSM (10 mM MgCl$_2$, 500 mM Sucrose in 50 mM Tris, pH 7.5) for 30 min at 37 °C. After centrifugation at 4,000 rpm (~3,200 × g) for 10 min, the supernatant was collected as the cell wall fraction. The pellet was washed gently with TSM and re-suspended in 2 u/ml DNase, 2 μg/ml RNase, 1× Protease Inhibitor in Lysis buffer. The protoplasts were lysed by sonication (Branson SFX250 Sonifier, microtip, 40% maximum power, 10 s, 8 cycles). After removing DNA by incubating for 30 min at 37 °C, the cell lysate was subjected to ultracentrifugation at 40,000 rpm (~86,000 × g), 1 h. The supernatant was collected as the cytoplasmic fraction. Proteins in the cell wall and cytoplasmic fractions were concentrated by TCA precipitation. The pellet was washed with the Lysis buffer, fully solubilized with 1× SDS sample buffer, and saved as the membrane fraction.

**Coomassie staining and Immunoblotting**. To examine the exoprotein profile, proteins in the culture supernatant were separated on 12% SDS-PAGE gels and stained with Coomassie Brilliant Blue.

For immunoblotting, proteins were transferred onto a nitrocellulose membrane. The membrane was blocked with 5% milk, probed with the indicated primary antibody, and incubated with Alexa Fluor 680-conjugated goat anti-rabbit or anti-mouse IgG (Invitrogen, 1:25,000) as a secondary antibody. Primary antibodies used in this study were rabbit anti-LukA[20] (1:5,000), rabbit anti-LukB[20] (1:1,000), rabbit anti-LukE (for detecting the S-subunit of other leukocidins)[84] (1:5,000), rabbit anti-LukD (for detecting the F-subunit of other leukocidins)[85] (1:7,500), rabbit anti-α-toxin (Sigma, 1:5,000), rabbit anti-IsdA[86] (1:25,000), rabbit anti-sortase A[86] (1:20,000), rabbit anti-SaeR[87] (1:2,000), mouse anti-His (Cell Sciences, 1:1,000), and mouse anti-LukAB mix (CC8-1-4.3.1.2.5.3 + CC30-3-10.1.5.9 + CC45-1-11.3.5, 1 μg/ml each). The monoclonal anti-LukAB antibodies were custom-made at Envigo with standard procedures for generating mouse monoclonal hybridoma. Due to high sequence similarity, we used a cross-reactive antibody to probe the respective S- or F-subunit of HlgAB, HlgCB, LukED, and PVL[85]. Images were acquired with the Odyssey Clx imaging system (Li-Cor Biosciences). Quantification of protein signals was performed using the Western analysis with the Image Studio software (Li-Cor Biosciences). Protein signals were normalized to WT or a purified recombinant protein control on each membrane.

**Isolation of bacteria from mice**. All experiments involving animals were reviewed and approved by the Institutional Animal Care and Use Committee of NYU Langone Health and were performed according to guidelines from the National Institutes of Health (NIH), the Animal Welfare Act, and US Federal Law. Mice were housed in specific pathogen-free facilities and male C57BL/6 J mice at 6-15 weeks of age were randomly assigned to infection groups.

Mice were infected intraperitoneally with 300 μl USA300 WT ($1.1 \times 10^8$ CFU) or ΔlukAB ($1.4 \times 10^8$ CFU). After ~20 h of infection, mice were euthanized and the peritoneal cavity was lavaged 3 times with 5 ml PBS each time. The peritoneal lavage fluid was centrifuged at 4,000 rpm (~3,200 × g), 30 min, 4 °C. The murine cells were further lysed with 0.1 % saponin, 1x Protease Inhibitor in 1 ml PBS for 30 min on ice. The bacterial pellet was collected after centrifugation at 14,000 rpm (~21,000 × g), 20 min at 4 °C. The pellet was lysed and the level of LukAB was examined by immunoblots as described above. Bacterial CFU was enumerated by serial dilutions in PBS and plating on TSA.

**Intoxication assays**. Primary human polymorphonuclear leukocytes (PMNs) from anonymous, healthy donors (New York Blood Center) were isolated from buffy coats as previously described[88]. Briefly, erythrocytes were removed by incubating with 0.9% sodium chloride, 3% Dextran 500 at room temperature for 25 min. The top fraction containing the PMNs and peripheral blood mononuclear cells (PBMCs) was washed and separated by centrifuging with Ficoll (Ficoll-Paque PLUS, Cytiva) at 805 × g for 30 min without brake. The pellet was collected as PMNs and further lysed with ACK lysing buffer (Gibco) to remove contaminating erythrocytes. The cells were re-suspended in PMN medium (0.1% human serum albumin, 10 mM HEPES in RPMI) and filtered through a 70-μm nylon cell strainer before use.

To prepare the bacterial lysate for PMN intoxication, bacteria were washed, normalized to $1 \times 10^9$ CFU/ml, and lysed with 40 μg/ml lysostaphin in PMN medium. Bacteria were also incubated with PMN medium only as a negative control. Bacterial debris were removed by centrifugation and the supernatant was filtered and saved as the bacterial lysate.

In each well of a tissue culture-treated flat-bottom 96-well plate, $2 \times 10^5$ PMNs and bacterial lysate equivalent to $2 \times 10^6$ CFU were mixed. When appropriate, 2.5 μg/ml affinity-purified anti-LukA[22] was included. After incubation at 37 °C in 5% CO$_2$ for 2 h, PMN viability was evaluated using the CytoTox-ONE Homogeneous Membrane Integrity Assay (Promega). In brief, 25 μl of the supernatant was mixed with 25 μl of the LDH reagent and incubated for 15 min at room temperature. Fluorescence (Excitation 560 nm; Emission 590 nm) was measured using a PerkinElmer 2103 Envision multilabel plate reader and

normalized to wells containing PMNs only (0% cell lysis) and PMNs with Triton X-100 (100% cell lysis).

**Infection assays**. Before infecting PMNs, OD600-normalized bacteria were washed with PBS and incubated with 100 μg/ml chloramphenicol, 40 μg/ml tetra-cycline, 1% ethanol (vehicle control for chloramphenicol), or 0.5% ethanol-0.5% methanol (vehicle control for tetracycline) on ice. Tetracycline was used for hemin-inducible *lukAB* due to the presence of the chloramphenicol resistant plasmid. After at least 30 min treatment, 20 μl of bacteria were aliquoted into each well of a 96-well plate, followed by 40 μl of PMN medium with antibiotics or vehicle control at 2x concentration and 40 μl PMNs at $5 \times 10^6$ cells/ml. The mixture was incubated at 37 °C in 5% CO$_2$ for 1 h and PMN viability was measured as described above.

**Effect of antibiotics**. To evaluate the effect of antibiotics on bacterial growth, the assays were set up in the same way as the infection assay except the bacteria were incubated for 2 h to allow for significant bacterial growth without antibiotics. The CFU of the input bacteria and after 2 h incubation was enumerated by serial dilution and plating on TSA plates.

XylE reporter assay was performed as described previously[32]. Bacteria were grown to the early stationary phase in TSB with 5 μg/ml chloramphenicol. Washed bacteria normalized to $1 \times 10^9$ CFU/ml were incubated with 40 μg/ml tetracycline or 0.5% ethanol-0.5% methanol (vehicle control) for 30 min on ice. In a 24-well plate, 0.4 ml of bacteria was diluted with 1.6 ml of PMN medium with or without 40 μg/ml tetracycline to mimic the bacteria concentration used in the PMN infection assays. Hemin was added into the culture to a final concentration of 0, 1, 2, 4, or 8 μM. After 1 h incubation at 37 °C in 5% CO$_2$, bacteria were pelleted by centrifugation at 13,000 rpm (~16,000 × g), 2 min. The pellet was washed with 20 mM potassium phosphate (pH 8.0) and re-suspended in 100 μl of 10% acetone, 20 μg/ml lysostaphin in 100 mM potassium phosphate (pH 8.0). After incubation at 37 °C with shaking for 20 min and then incubated on ice for 5 min, cell debris was removed by centrifugation at 14,000 rpm (~21,000 × g), 30 min at 4 °C. In a flat bottom 96-well plate, 20 μl of the supernatant was mixed with 200 μl of 0.2 mM pyrocatechol in 100 mM potassium phosphate (pH 8.0). Absorbance at 375 nm was measured immediately and then every 5 min for 30 min using a PerkinElmer 2103 Envision multilabel plate reader. Total protein concentration was determined using the Pierce BCA Protein Assay (Thermo Scientific). The rate of 2-hydroxymuconic semialdehyde formation (XylE activity) was calculated using linear regression as a function of time. The relative reporter activity was presented as fold-change of XylE enzyme activity per milligram of total proteins compared to no promoter control.

**Immunofluorescence microscopy**. To visualize LukAB by immunofluorescence microscopy, washed bacterial cells were settled on a clean coverslip for 15 min and fixed with 4% paraformaldehyde for 15 min at room temperature. The coverslip was blocked with 100 μg/ml human IgG, 5% BSA in PBS, and then incubated with 10 μg/ml affinity-purified rabbit anti-LukA[22], mouse anti-FLAG (Sigma) (1:100), or rabbit anti-protein A (Sigma) (1:1,000) as a primary antibody in PBS containing 1% BSA. The coverslip was washed three times and further incubated in the dark with Alexa Fluor 594-conjugated goat anti-rabbit or anti-mouse (Abcam) (1:1,000) in PBS with 1% BSA. Each of the incubation steps was done at room temperature for 1 h. The coverslip was then washed six times and stained in the dark with 1 μg/ml BODIPY FL vancomycin (Invitrogen) in PBS for 10 min. The coverslip was washed three times, mounted with Fluoromount G (SouthernBiotech) and air-dried overnight in dark.

The SNAP-tagged LukAB was labeled with 20 μM SNAP-Surface 594 (NEB) in PBS containing 0.5% BSA for 30 min at 37 °C. The cells were washed, fixed with 4% paraformaldehyde for 15 min, stained with 1 μg/ml BODIPY FL vancomycin (Invitrogen) in PBS for 10 min, and loaded on 1.5% agarose pads for imaging.

To expose LukAB epitopes inside the cell wall, we adapted a lysostaphin treatment method described previously[89]. Fixed bacteria on a coverslip were treated with 10 μg/ml lysostaphin in TSM for 15 min at 37 °C. The blocking and antibody staining steps were performed as described above except PBS was replaced by TSM in all steps. After washing off the unbound secondary antibody, the coverslip was incubated in dark with 10 μg/ml FM 1-43 (Invitrogen) for 20 min at room temperature. To evaluate the effect of lysostaphin treatment, the coverslip was stained sequentially with 1 μg/ml BODIPY FL vancomycin and 10 μg/ml FM 4-64 at room temperature. The coverslip was mounted with Fluoromount G and air-dried overnight in dark.

Slides were imaged using Plan-Apochromat 63x/1.4 Oil DIC M27 Elyra objective on a Zeiss 880 Laser Scanning Confocal microscope with Airyscan. BODIPY FL and FM 1-43 were imaged using 488 nm excitation and 495-550 nm emission filters. Alexa Fluor 594, SNAP-surface 594 and mCherry were imaged using 594 nm excitation and 605 nm long pass emission filter. FM 4-64 were imaged using 488 nm excitation and 605 nm long pass emission filter. Z-stacks of fluorescent channels were collected at 0.17 μm steps to cover the depth of the bacteria. 3D Airyscan processing was performed for raw images using the Zen software (Zeiss) with automatic strength. A single slice of brightfield was captured as a reference for bacteria position. Identical settings were applied to all samples in each experiment. For Structured Illumination Microscopy (SIM), samples were imaged using an SR Apo TIRF AC 100xH objective on a N-SIM S Super Resolution

Microscope and images were reconstructed using the NIS-elements AR software (Nikon).

All images were processed in the Fiji distribution of the ImageJ software[90,91]. Stack images were Z-projected by maximum intensity. Fluorescence profile was plotted by drawing a line anti-clockwise from the top along the cell wall or membrane staining and measuring the fluorescence intensity. The relative distance to the septum on the 2D projected images was defined as the ratio of the shortest path between the LukAB foci and septum along the cell wall versus half of the length of the hemisphere containing the foci. Quantifications of LukAB foci were performed using the ImageJ plugin MicrobeJ[92]. Bacteria cells that were intact and had clear separation with other cells were identified automatically based on the membrane or cell wall staining with manual corrections. LukAB foci associated with each cell were identified as point maxima and the foci between two cells were assigned to one cell automatically by the software. For analyzing LukAB distribution in the *mprF* and *ypfP* mutants, only isolated single cells were used for quantification. The threshold for maxima detection was set based on the mutant and secondary antibody-only control in each experiment. A list of foci with fluorescence intensity and parental bacteria information was generated by MicrobeJ and further analyzed in MATLAB and Excel.

**Cryo-immunogold electron microscopy.** Bacteria were grown to early stationary phase (5 h), washed 3× with PBS at 2000 × g for 10 min and fixed with 3.2% paraformaldehyde + 0.0125% or 0.05% glutaraldehyde for 30 min. After another wash in PBS, the bacteria were harvested, gelatin-embedded and infiltrated with 2.3 M Sucrose according to the method described previously[93]. Ultrathin sections were cut at −110 °C with an RMC MTX/CRX cryo-ultramicrotome (Boeckeler Instruments Inc., Tucson AZ, USA) transferred to carbon- and pioloform-coated EM-grids and blocked with 0.3% BSA, 0.01 M Glycin, 3% CWFG in PBS. The sections were incubated with appropriate dilutions of affinity-purified rabbit polyclonal anti-LukA[22] or mouse monoclonal anti-FLAG in the same buffer. Secondary antibody-incubations were carried out with goat anti-rabbit and goat anti-mouse antibodies coupled to 12 nm or 6 nm gold particles (Jackson ImmunoResearch). Specimens were then contrasted and embedded with uranyl-acetate/methyl-cellulose following the method described[94] and analyzed in a Leo 912AB (or Leo 906) transmission electron microscope operated at 120 kV (or 100 kV) (Zeiss, Oberkochen, Germany). Micrograph-mosaics were scanned using a bottom mount Cantega (or sidemount Morada) digital camera (SIS, Münster, Germany) with ImageSP software from TRS (Tröndle, Moorenweis, Germany).

**Dot blot screen on the Nebraska Tn mutant library.** Mutants in the Nebraska transposon mutant library were grown in 96-well plates for 24 h. Control strains (JE2 WT, *lukA::bursa*, *spa::bursa*, *saeR::bursa*), purified recombinant LukAB, and an empty well were included in each plate. Washed bacterial pellets were re-suspended and diluted 1:4 with 1× SDS sample buffer (2% SDS, 10% glycerol, 147 mM 2-mercaptoethanol, 12.5 mM EDTA, 0.02% bromophenol blue in 50 mM Tris, pH 6.8). After 30 min incubation at room temperature, the plate was centrifuged at 4,000 rpm (~3,200 × g), 10 min, and 100 µl of the supernatant was loaded onto a nitrocellulose membrane through the Bio-Dot microfiltration apparatus as per manufacture instructions (Bio-Rad). The membrane was removed after the samples were filtered through the membrane and washed twice with PBST (0.1% Tween 20 in PBS). The membrane was blocked with 5% milk, 2 µg/ml human IgG in PBST at 4 °C for overnight, followed by incubation with a rabbit anti-LukA antibody (1:5,000) as the primary antibody and an Alexa Fluor 680 goat anti-rabbit antibody (1:25,000) as the secondary antibody at room temperature for 1.5 h each. The membrane was scanned with the Odyssey Clx imaging system (Li-Cor Biosciences). Quantification of protein signals was performed using the Grid analysis in the Image Studio software (Li-Cor Biosciences) and the relative LukA abundance was calculated for each mutant against control strains.

After the primary screen with 1,920 mutants in the library, 161 mutants with LukA signal higher than 150% and 121 mutants with lower than 50% of the WT signal were picked for a secondary screen. The dot blot measurement of LukAB levels was repeated three times for these mutants. From the secondary screen, 21 mutants with higher and lower LukA signal (total 42 mutants) were picked and examined for LukAB, LukF, and α-toxin levels by immunoblotting. Mutants that only influence LukAB but not LukF or α-toxin levels were selected for further examination, including transducing the mutation to the LAC WT strain and complementing the phenotype.

**Cytochrome c binding assay.** The positive surface charge was determined based on bacterial ability to repulse cationic protein cytochrome c as described previously[95,96]. Early stationary phase bacteria were adjusted to OD600 of ~1.1. Aliquots of 2 ml bacteria were washed twice with sodium acetate buffer (20 mM, pH 4.6). The pellet was re-suspended in 0.5 ml 0.25 mg/ml cytochrome c in sodium acetate buffer and incubated with shaking for 15 min at 37 °C. The bacteria were centrifuged at 13,000 rpm (~16,000 × g), 2 min, the supernatant was aliquoted and the absorbance at 410 nm was measured using a PerkinElmer 2103 Envision multilabel plate reader. The measurement was normalized to 0.25 mg/ml cytochrome c as the percentage of unbound cytochrome c.

**Hydrophobicity assay.** Surface hydrophobicity was measured as bacterial affinity to organic solvents as described previously[48]. Early stationary phase bacteria were washed with PBS and adjusted to OD600 of ~0.5. Aliquots of 3 ml bacteria were mixed with 50 µl dodecane vigorously by vortexing for 1 min. The phases were allowed to separate by leaving the tube still for 10 min. The OD600 of the PBS phase was measured and normalized to WT as relative surface hydrophobicity.

**GFP reporter assay.** Strains containing the GFP reporter plasmids were grown to the early stationary phase. Bacteria were washed and diluted 1:2 with PBS. The GFP fluorescence and OD600 in 200 µl of the suspension were measured using a PerkinElmer EnVision 2103 Multilabel Reader. The GFP signal was normalized by the OD600 readings.

**Quantitative mass spectrometry.** After bacteria were grown for 5 h or 24 h, washed bacterial pellets were normalized and re-suspended with 1× SDS sample buffer (2% SDS, 10% glycerol, 147 mM 2-mercaptoethanol, 12.5 mM EDTA, 0.02% bromophenol blue in 50 mM Tris, pH 6.8). The suspension was incubated for 30 min at room temperature before centrifuging at 4,000 rpm (~3,200 × g), 10 min. The supernatant was filtered through a 0.22 µm filter and stored at −80 °C. A portion of the sample was boiled, separated by SDS-PAGE, stained with SYPRO Ruby Protein Gel Stain (Invitrogen), and visualized with the ChemiDoc imager (Bio-Rad).

For mass spectrometry (MS), the protein isolates (triplicate preparations) were reduced with 0.02 M dithiothreitol and alkylated with 0.05 M iodoacetamide. To remove detergent and other non-MS compatible components, the samples were run approximately 2 cm into a NuPAGE gel (LifeTechnologies) so that the entire sample was concentrated into a single gel band. The gel was stained with GelCode Blue Stain Reagent (Thermo) and the band containing all proteins was excised and destained for 15 min in a 1:1 (v/v) solution of methanol and 100 mM ammonium bicarbonate. The destaining solution was removed, another aliquot of fresh destaining solution added, and the samples were destained for another 15 min. This was repeated for another 4 cycles. The gel bands were dehydrated by washing with acetonitrile, and then further dried by placing them in a SpeedVac for 20 min. 300 ng of sequencing grade modified trypsin (Promega) was added directly to the dried gel followed by 100 µl of 100 mM ammonium bicarbonate to cover the gel bands. The digestion proceeded overnight with gentle agitation at room temperature. The resulting peptides were extracted and desalted as previously described[97].

An aliquot of each sample was loaded onto an Acclaim PepMap trap column (75 µm × 2 cm, C18, 3 µm, 100 Å) in line with an EASY-Spray analytical column (50 cm × 75 µm ID PepMap C18, 2 µm bead size) using the auto sampler of an EASY-nLC 1000 HPLC (Thermo Fisher Scientific) with solvent A consisting of 2% acetonitrile in 0.5% acetic acid and solvent B consisting of 80% acetonitrile in 0.5% acetic acid. The peptides were gradient eluted into an Orbitrap Fusion Lumos mass spectrometer (Thermo Fisher Scientific) using the following gradient: 5–20% in 120 min, 20–40% in 20 min, followed by 40–100% in 10 min. MS1 spectra were recorded with a resolution of 70,000 (at m/z 200), an AGC target of 1e6, with a maximum ion time of 120 ms, and a scan range from 400 to 1500 m/z. The MS/MS spectra were collected using the top 20 method with a resolution of 17,500 (at m/z 200), an AGC target of 5e4, maximum ion time of 120 ms, one microscan, 2 m/z isolation window, a Normalized Collision Energy (NCE) of 27, excluded ions of charge state +1 and +5, and a dynamic exclusion of 30 s.

Spectra were searched against a Uniprot *Staphylococcus aureus* USA300, USA300 TCH959, and Newman combined database using the MaxQuant software suite (version 1.5.2.8). For the first search the peptide tolerance was set to 20 ppm and for the main search peptide tolerance was 4.5 ppm. Trypsin specific cleavage was selected with 2 missed cleavages. A peptide spectrum match (PSM) FDR of 1% and a Protein FDR of 1% was selected for identification. Carbamidomethylation of Cys was added as a static modification. Oxidation of Met, deamidation of Asn and Gln and acetylation of the protein N terminus were the allowed variable modifications. A filter was applied to select proteins with more than ten PSMs in at least two samples in each condition and data were further analyzed with MATLAB. Venn diagrams were generated using the Venn function from MATLAB file exchange. Statistical differences in WT samples between 5 h and 24 h were determined by t tests on PSMs. The heatmap containing samples with a *p*-value ≤ 0.05 was generated by showing the z-scores of all values and the unsupervised hierarchical clustering of protein species and samples.

The exoprotein PSM data were obtained from a previous in-house study[27] (MassIVE ID MSV000080260) where the bacteria strain and the growth conditions were identical to the new study described here. Of note, this comparison highlights that the relative proteome composition is different in the culture supernatant from bacterial surface. The study was not aimed to compare the abundance of a single protein between the two compartments as these were different experiments.

**Bioinformatics predictions.** The signal sequences of leukocidins were predicted using SignalP-5.0[51]. The predicted signal sequences and mature protein sequences were aligned using Clustal Omega (EMEL-EBI) with default settings[98]. The localization of proteins was predicted using PSORTb-3.0.2[50] and assigned to each of the proteins identified in the mass spectrometry. To search for known motifs that

mediate cell surface anchoring in LukAB, the homology search was performed for LukA and LukB using the ScanProsite tool[39] and the HMMER web server[38].

**Statistical analysis**. Except for the mass spectrometry studies, GraphPad Prism 8 was used for all statistical analyses. All statistical details of experiments can be found in the figure legends.

**Reporting summary**. Further information on research design is available in the Nature Research Reporting Summary linked to this article.

## Data availability

The mass spectrometric raw data generated in this study are accessible in the MassIVE database under ID: MSV000086238 (https://doi.org/10.25345/C5NF5S). The exoprotein mass spectrometry data were obtained from the MassIVE database under ID: MSV000080260. The nucleotide sequences of genes used this manuscript were acquired from NCBI SAUSA300_FPR3757 genome (NC_007793.1). All other data supporting the key findings of this study are available from the corresponding author upon reasonable request. Source data are provided with this paper.

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

## Acknowledgements

We thank members of the Torres laboratory for insightful discussions and comments on this manuscript, and especially Dr. Andrew Perault and Dr. Erin Zwack for commenting on the revised manuscript. We thank Dr. Yan Deng and Michael Cammer (Microscopy Laboratory, New York University Grossman School of Medicine) for assistance in performing the microscopy experiments. We thank Dr. Kayan Tam for generating the monoclonal antibodies. We thank Dr. Eric Skaar for the gift of sortase A and IsdA antibodies and Dr. Taeok Bae for the gift of SaeR antibodies. The Nebraska transposon mutant library was obtained from the Network of Antimicrobial Resistance in *S. aureus* program, which was supported under National Institutes of Health (NIH)-National Institute of Allergy and Infectious Diseases contract HHSN272200700055C. We thank BEI Resources for providing individual transposon mutants. This work was supported in part by the NIH-National Institute of Allergy and Infectious Diseases award numbers R01s AI099394, AI105129, AI140754, and HHSN272201400019C to V.J.T., the Max Planck Society to G.M., and the Cystic Fibrosis Postdoctoral Research Fellowship Award LACEY19FO to K.A.L. The NYU Langone Health Microscopy Laboratory and the Proteomics Laboratory are partially supported by the Cancer Center Support Grant P30CA016087 from the NIH-National Cancer Institute. The mass spectrometric

experiments were also supported with a shared instrumentation grant from the NIH, 1S10OD010582 for the purchase of an Orbitrap Fusion Lumos. V.J.T. is a Burroughs Wellcome Fund Investigator in the pathogenesis of infectious diseases.

## Author contributions

X.Z. and V.J.T. designed the study. X.Z. performed experiments. G.M. and C.G. performed the TEM. K.A.L. performed mouse infections. J.R.C. and B.M.U. performed mass spectrometry. X.Z. and V.J.T. wrote the manuscript, and all authors commented on the manuscript.

## Competing interests

V.J.T. is an inventor on patents and patent applications filed by New York University, which are currently under commercial license to Janssen Biotech Inc. (patent nos. US10669329; US10202440; US10087243; MY-175062-A; BR112013032774-0; 60 2014 068 192.1; 40000719B; 1190640B9; 10-2050267; 2012273125; 2012273123; 10,781,246; 10,301,378; 9,783,597; 9,657,103; 9,644,023; 9,481,723; 9,480,726; 9,091,689; 8,846,609; 6758363; 6452765; 6,170,913; 6,093,760; 3441474; 3403669; 3011012; 2,720,754; 2720714; 2,635,462; 2,613,135; 2,609,650; 730359; 710,439; 619,942; 619,938; 357,938; 343,589; 340,446; and 229922). Janssen Biotech Inc. provides research funding and other payments associated with the licensing agreement. All other authors declare no conflicts of interest.
