## [Peer Review File · Nature Communications]

The cell envelope of *Staphylococcus aureus* selectively controls the sorting of virulence factorsREVIEWER COMMENTS

Reviewer #1 (Remarks to the Author):

In the report, the cell envelope of *Staphylococcus aureus* selectively controls the secretion of virulence factors, Zheng and colleagues show that LukAB toxins localize in a reservoir in the cell envelope of *S. aureus* cells. They provide genetic evidence of the mechanism involved in LukAB deployment in the cell envelope. They show other secreted proteins to follow a similar path, arguing that this may represent a multistep secretion system that allow bacteria to generate an exoprotein storage in bacterial cell wall to allow fast response during infections. This is a potentially interesting manuscript. I have comments to specific experiments and controls.

Fig. 1D. Detection of LukA by immunoblotting. Detection of LukB is missing. In panel 1C, the authors detected the two toxins LukA and LukB. Is there a reason why Fig 1D shows only LukA?

The section in page 5 needs a revision. I read it several times and I am still not sure I understood Fig. 2B. In this section, the authors investigated the role of bacterial-associated LukAB by inhibiting bacterial protein synthesis using antibiotics and cocultured with PMN. They showed that, infecting PMN with antibiotics reduced the cytotoxicity of *S. aureus*. It is difficult to see the connection of this result to LukAB expression. I think the authors tried to show that lukAB expression is induced during PMN infection. Antibiotics inhibit LukAB translation and prevented the LukAB production during infection, causing a reduction in *S. aureus* cytotoxicity. I still don't understand the point of this experiment.

To decouple gene expression and protein production, the authors engineered an inducible strain and showed that, upon artificial lukAB induction, *S. aureus* killed PMN with or without antibiotics. Here, I think the authors tried to demonstrate that the presence of antibiotic in the killing medium only inhibits the production of new LukAB proteins; the old LukAB protein already attached to the cell wall proved enough to kill PMN. As I said, I'm not sure I understood. Moreover, additional controls are needed to show that LukAB contributes to the killing effect. For instance, mutants in the other leukocidins, which do not accumulate in the cell wall, should not produce the PMN killing effect detected in Fig 2B-D.

Is it possible to use purified cell wall from WT (which contains LukAB) or maybe purified LukAB to complement DlukAB -PMN interaction? This will probe that LukAB is required for the USA300-PMN and will facilitate to understand this section.

I have two comments in relation to the fluorescent microscopy experiment presented in page 6. 1) The authors performed immunostaining experiments to localize LukAB signal in *S. aureus* cells. It is well known that *S. aureus* immunostaining produces artifacts as the antibodies bind non-specifically to the staphylococcal cell surface adhesins. To avoid this, the authors performed the experiments in a Dspa Dsbi mutant, which lacks two of the most relevant adhesins. The authors show localization of LukAB in large, round foci, which is also the classical signal distribution pattern of non-specific binding. The problem here is that, whether the antibodies bind to LukAB or to remaining cell-exposed adhesins, they may generate the same signal distribution pattern and it would be difficult to determine if it is a legitimate signal.

Two suggestions below to help authors to overcome this issue and to complement their fluorescence microscopy experiments.

The authors should add a supplemental figure in which the signal of the control strain (Dspa Dsbi) is shown at different exposure times and compared to that of the sample strain. In the current stage, the Alexa Fluor 594 channel produces a black field. It would be interesting to show images of this channel at different exposure times, to reach the exposure level in which the background signal allows the visualization of the cell shape and still no foci can be distinguished. The signal should be different in the sample strain, in which different exposure times should show cell wall foci in all cases.

Another option is to obtain a similar signal distribution pattern using another approach that does

not involve antibodies. This would validate the signal obtained using antibodies. I recommend the use of SNAP or HALO tags to LukAB. These tags bind fluorophores very specifically and the signal can be traced using the Alexa fluor 594 channel. No antibodies involved. A supplemental figure showing that immunostaining and SNAP staining produce similar signal distribution would be sufficient to validate Figure 3.

2) My second comment is related to the use of protoplasts. How can the authors be sure that cells were protoplast, that no cell wall remains in the cells? It is possible to follow the formation of protoplast in rod-shaped bacteria with the change of shape to spheres but in the case of the already spherical *S. aureus*, how could you know that no cell wall remains in the cells?

The authors performed a genetic screen and identified the *mprF* and *ypfP* genes involved in LukAB cell wall deposition and secretion. These experiments are very interesting. I just have one suggestion to the authors. MprF adds lisil residues to PG. *ypfP* contributes to the LTA formation, which bind alanyl residues and Mg²⁺ cations. Therefore, both *mprF* and *ypfP* genes contribute a net positive charge to the *S. aureus* surface envelope by lysinylating PG or alanylating teichoic acids. Possibly, LukAB (as well as the other proteins) is retained in the cell wall as a consequence of the net positive charges that LPG or LTA add to the cell wall. If this is the case, adding Mg²⁺ to the culture medium will add extra net positive charge to the cell wall thus it will complement DmprF and DypfP mutants in their capacity to retain LukAB in the cell wall.

Reviewer #2 (Remarks to the Author):

In this study, the authors address how secretion of the leukocidin LukAB, which is known to be both secreted and cell envelope-associated, is controlled by *S. aureus*. *S. aureus* secretes numerous factors important both in its physiology and in virulence. Although how proteins are secreted through the membrane is generally known, there are major gaps in our understanding of how proteins move from the cell membrane to other sites in the cell envelope or beyond. The authors are addressing a fundamental question, the experiments in the manuscript are well done, and I believe that this manuscript will find a receptive audience in those interested in this problem no matter where it is published. However, I do not know whether this manuscript meets the bar for publication in Nature Communications because it still does not provide a mechanism-based understanding. I will leave that up to the editor because I wouldn't mind seeing it published there.

The key findings of this study are the following: 1) Cell-envelope-associated LukAB is an active toxin. 2) LukAB forms a punctate foci pattern at the cell envelope. 3) LukAB is found in two distinct cell envelope "depots" (surface-exposed and membrane-proximal). 4) LukAB secretion and localization are altered in deletion mutants of two genes, *mprF* and *ypfP*, that encode cell envelope biogenesis enzymes. 5) Other proteins known to be secreted by *S. aureus* are also found in two distinct "depots" and are affected by deletion of *mprF* and *ypfP*.

Major Comments:

The authors frequently discuss the idea of a "multistep process" for LukAB secretion in which LukAB, after translocation across the membrane, is first stored in a membrane-proximal compartment and then a surface-exposed compartment before finally becoming fully dissociated from the cell. While the evidence does clearly suggest that there are two discrete cell envelope-associated "depots" where LukAB is found, evidence for a "multistep process" is lacking. For example, it is unclear whether individual LukAB dimers traverse from the membrane-proximal compartment to the surface-exposed compartment (rather than from the membrane-proximal compartment straight to the supernatant). Similarly, whether or not all secreted LukAB dimers initially spend any significant amount of time in either compartment is unknown.

In the results from the targeted mutant screen, it is clear that deletions of *mprF* and *ypfP* cause defective secretion of LukAB. While *mprF* was found as a hit in the Nebraska library screen, *ypfP* was not. Is there a reason that the *ypfP* mutant did not come up in this screen? Additionally, the

ItaA mutant was observed as a hit for increased LukAB secretion in the Nebraska library screen, which is confusing because LtaA is the flippase that flips the LTA glycolipid to the outer leaflet of the membrane after it is generated by YpfP on the inner leaflet of the membrane. For this reason, YpfP-LtaA should not be referred to as a single entity. Is there a way to reconcile opposing effects of ypfP and ItaA deletion on LukAB secretion given that they play roles in adjacent steps in LTA biogenesis? Do the authors call it YpfP-LtaA because they can only complement with the full operon? (The ypfP deletion may have polar effects on ItaA.) They should clarify, and they need to deal with the ItaA issue in some manner. Also related to the two screens, the authors should clarify in the main text the process they went through to narrow down the initial hits. The Methods section titled "Dot blot screen on the Nebraska library" has a very informative paragraph, and it would be helpful if this information was present in the main text. There are still some aspects that are not clear however. For example, what are the hits found in Figure S6A? There are more than 21 in each set of hits (for higher and lower signal), though in the Methods section, it says that "21 mutants with most and least LukA signal (total 42 mutants)" were examined? Are these Figure S6A hits a subset of the "161 mutants with LukA signal with higher than 150% and 121 mutants with lower than 50% of the WT signal"? If so, why are these displayed but not the others? There should also be information about the targeted screen - why were these genes chosen, and why was esaA looked at in the LAC background? In general, for mutants discarded because they are "known" or have "potential" to regulate lukAB transcription, what evidence is there that lukAB transcription is affected for these mutants (for the ones with "potential" to regulate, specifically)? It would be beneficial to have a table showing results from all the mutants and why they were or were not discarded for future analysis.

Minor Comments/Questions:

General Comments/Questions

- The term "bacteria-associated" should be replaced with "cell envelope-associated" to be more descriptive.
- The title should better reflect the specific findings of the paper (something about sorting into two distinct cell envelope compartments, perhaps).

Abstract and Introduction:

- In the Abstract, "anchored to" in the sentence beginning "Intriguingly, one of the leukocidins..." should be changed to "associated with" or something similar, as "anchored" suggests a covalent attachment.
- In the Abstract the phrase "ready to be deployed" should be removed, as the manuscript does not provide evidence of a "deployment" mechanism or trigger; it could just be that cell envelope-associated LukAB is gradually shed by the cell rather than being specifically "deployed."
- Remove "to date" in the first paragraph; this is redundant with "ongoing discussion."
- Clarify what is known about the component(s) of the cell envelope that GW repeats and LysM domains interact with.

The introduction ends with the statement that the results here "establish the role of the *S. aureus* cell envelope for the sorting and secretion of selective exoproteins." Other papers have commented on involvement of cell envelope factors such as WTA in "spatial regulation" of secreted proteins such as Atl (DOI: 10.1128/AAC.00323-18; and WTA has also been implicated in spatial regulation of membrane-anchored proteins by an unknown mechanism; the authors' own data also supports the previous evidence that WTA is somehow involved in sorting/secretion). In addition, the membrane protein LyrA (now called SpdC) was also implicated in protein secretion (and so were other "surface protein display" factors). The glucosaminidase SagB, which affects PG strand length, was independently implicated in protein secretion. (A very recent Nat Micro paper shows that LyrA/SpdC regulates the activity of SagB, so the two are connected.) Therefore, it was known previously that the *S. aureus* cell envelope is important for sorting and secretion of exoproteins, and the authors do in part acknowledge this elsewhere. So while this manuscript is an interesting, and I think useful, contribution to the growing literature, the last sentence of the introduction overstates the findings. It would be good to rephrase for clarity and perhaps cite previous literature.

Comments on the section titled "LukAB is the only leukocidin associated with the bacterial cell."

- The title should be changed, as this was already known; the novel findings are the temporal data, the finding that the abnormal localization of LukAB is not due to heterodimer formation, the subcellular localization of LukAB, and the finding that bacterial-associated LukAB is observed in vivo.
- The fact that LukS* recognizes the S subunits of the other leukocidins but not that of LukAB suggests (same with LukF* for the F subunits) that there are important differences between LukAB and these other leukocidins at the biochemical level. A comment on notable differences is warranted (if known). It would also be nice to clarify in the figure captions that the other leukocidins run at the same MW as each other, explaining why there's only one apparent band in the LukS* and LukF* blots.
- The following is definitely not necessary to address but would be interesting if there is remaining sample that can be analyzed without doing another mouse experiment: the other leukocidins are shown to not be bacterially associated in vitro - is this true in vivo as well?

Comments on the section titled "Bacteria-associated LukAB contributes to the cytotoxicity of USA300."

- Is it known whether the LukAB dimer is required for cytotoxicity, or does each individual protein have some effect on its own?
- For Figure 2A, is Anti-LukA negative sera available as a control to ensure that other sera components are not responsible for decreasing cytotoxicity?
- For Figures 2B and 2C, are there non-protein synthesis inhibitors that can be used to differentiate between the effects of just stopping protein synthesis and that of cell growth inhibition? Also, the amount of bacteria added in these assays after inhibitor treatment should be normalized by OD or cfus instead of by volume to control for growth inhibition during antibiotic incubation.

Comments on the section titled "LukAB forms discrete foci on USA300 cells."

- It would be nice to see a control in the experiment for Figure 3A where localization is examined with the anti-LukB antibody to ensure that the same pattern is observed. It would also be interesting to know if these foci still form if only LukA or only LukB is present.
- It would be nice to show data analogous to that in Figure 3B but for exponential phase cells rather than late stationary phase cells, and exponential phase cells have more LukAB foci according to the data in Figure 3F.
- The captions for the histogram figures should state how many cells were analyzed per condition.
- What phase were the cells in Figure 3C grown to? If they were grown to stationary phase, it would be better to do this analysis in exponential phase as there would be more septum-containing cells. For the current data, it does not seem that there are enough cells (68) to establish a trend.
- It should be clarified for Figure 3H that the FLAG tag is on LukA.
- For the sentence beginning "To minimize unspecific binding of the antibodies..." it would be good to explain for general readership that spa and sbi encode two of the most highly abundant proteins on the *S. aureus* cell surface.
- It wasn't clear to me that Fig 3b added much that justifies inclusion in the main text.

Comments on the section titled "LukAB is secreted through a multistep process."

- For the terminology "SDS-resistant compartment" and "SDS-susceptible compartment," it seems that these phrases should be used in this section only and then, afterwards, only "membrane-proximal" and "surface-exposed" should be used which are more informative/clear.
- Not necessary but would be interesting: is there a way to look at the impact of the SDS-resistant fraction of LukAB on PMN killing?

Comments on the section titled "MprF and YpfP-LtaA contribute to LukAB secretion."

- It would be good to bring up the secondary screen and any validation of the screen in the results section.
- For Figure S6A, the "common" names of the genes should be listed with the gene IDs. A table may be a better way to show the information in this figure.
- Figures 5H and 5I show that mprF and ypfP mutants are somewhat more cytotoxic than wild-type cells. How does this fit with previous reports that mprF and ypfP strains (and other strains defective in glycolipid-anchored LTA) are less virulent than wild-type?

- In Figure S6B, what are the bands between the LukA and Hla bands that appear to be present in some of the samples (mprF, tatC, srtA, ypfP, mecA)? Are they LukA variants?
- In the text, for the sentence starting "Without YpfP or LtaA, *S. aureus*..." it would be good to mention what is meant by "abnormal LTA". For ypfP mutants, LTAs are much longer than normal and have a phosphatidylglycerol anchor rather than a diglucosyl-diacylglycerol anchor, and for ltaA mutants, LTAs are intermediate in length (compared to WT and ypfP) and have some phosphatidylglycerol anchors and some diglucosyl-diacylglycerol anchors.
- It would be good to mention that ypfP cells are larger than wild-type cells, as this could affect why the ypfP mutant seems to have increased exoprotein secretion as shown in Figure 5A.
- Fix grammar: "Mutants known or potential to regulate lukAB transcription were excluded." Citations to these mutants should also be provided at the end of the sentence. See also the comment in the "major comments" section.
- As a control, it would be nice to analyze inactive ypfP and mprF point mutants to confirm that it is the products of their enzymatic activities that influence LukAB localization. YpfP/UgtP has been proposed to have other activities related to protein-protein interactions.
- It would be interesting (although not necessary) to test LukAB localization in other strains with altered surface charge like a D-alanylation mutant or a WTA mutant. Also, cytochrome C is supposed to be sensitive to positive surface charge, not just surface charge and this should be clarified in the text with a citation to the paper that reported the method originally for *S. aureus* (Peschel 1999 JBC). It is obvious how mprF deletion reduces positive surface charge, but it's less obvious how ypfP deletion increases positive surface charge. Is it due to increased D-ala? Comment.
- The PMN killing assay isn't particularly convincing to me in terms of the claim that "the mprF and ypfP mutants exhibited increased LukAB-mediated killing of human PMNs compared to WT USA300"... The effect size seems quite small so even though there's statistical significance, I don't think it's all that important; also for the ypfP mutant, lukAB promoter activity is elevated, so could it be related to the fact that there's more lukAB around?

Comments on the section titled "Multistep secretion of other exoproteins."

- For Figure 6B, it would be nice to have a His-tagged control protein that is known to not be present in either the SDS-susceptible or SDS-resistant portions but is present in the supernatant-- perhaps a His tagged non-LukAB leukocidin?
- For the third panel in Figure 7, there should be longer LTA shown as present because ypfP deletion does not cause loss of LTA. When revising this panel's LTA, the two small circles indicating the glucose units should be removed to show loss of the glycolipid anchor. In the first panel, the negative charge inside the cell is really confusing because it looks like a negatively charged something is moving to the outside of the cell and somehow becoming positively charged. It would be better to show positively charge L-PG moving from the inside to the outside or to show two steps to convey synthesis and then translocation. Also, the figure suggests low levels of L-PG on the inside compared with the outside. Is this known? LTA contains both negative and positive charges, with the latter at least partially neutralizing the former. It is not clear what happens to the cell envelope in ypfP mutants that leads to increased positive charge, but it would seem necessary to somehow represent that there is a difference. This gets at what they actually learned about mechanism. Possibly the long LTAs retain proteins at/near the surface of the envelope, but it could also be that other things change in the ypfP mutant that explain the results.
- More on Fig 7: I am also not sure about calling this figure a model when there is no suggested mechanism for how mprF and ypfP-ltaA cause these differences in sorting/secretion (other than a charge-based mechanism for MprF). Without grappling with some kind of mechanism for the ypfP results – and then figuring out how to depict a possible mechanism – the third panel of this figure does not clarify anything even if fixed to show the presence of LTA.
- For Figure S9E, it seems that deletion of ypfP causes increased secretion of IssA. Is there evidence to show that this is solely due to a transcriptional effect as the text suggests?

Comments on Discussion

- Phrases like "rapid release of virulence factors" and "masked from host immune surveillance, while available to be fired rapidly in specific environments" should be avoided as these are not supported by the data in the paper and are just speculative claims. Alternatively, rephrase to clarify speculative from established claims.

Comments on Methods

- What protease inhibitor are they using?

Reviewer #3 (Remarks to the Author):

Zheng et al describe how modification of cell membrane lipids and of extracellular lipoteichoic (LTA) polymers affect the retention vs. release of the bi-component leucocidin LukAB and other secretory proteins in *S. aureus*. These are new, important findings as they may help to understand Gram-positive bacterial mechanisms to store secreted proteins in the cell envelope. LukAB is one of the critical *S. aureus* virulence factors, whose accumulation at the bacterial surface and targeted release may contribute to pathogenicity. The study is sound and well written. It raises a number of questions the authors may consider in a revised manuscript:

1. The findings raise the question which interactions may retain LukAB in the envelope and which mechanisms will lead to release. SDS can release the proteins but this is an unphysiologically harsh treatment. The crucial role of LPG and LTA suggests a role of electrostatic interactions, which have been reported to be the major forces altered by the MprF and YpfP-dependent reactions. The authors could test if addition of high salt concentrations, in particular of calcium or magnesium ions, which are known to interact with the phosphate groups of LPG and LTA can trigger the release. Alternatively, altered pH values of chaotropic solutes such as urea could be tested. Maybe such experiments could reveal a potential *in vivo*-related condition that would promote the release of LukAB from the cell envelope. Do the different retained proteins share a specific net charge or another property?
2. Inactivation of YpfP has been found to have different consequences in different *S. aureus* strains. It does not eliminate LTA but alters its amount and potentially polymer length. Do the authors have an idea what is altered in their mutant? Did mutations in other genes related to LTA biosynthesis have an impact on LukAB retention (e.g. LTA polymerase LtaS or LTA-modifying enzymes DltABCD)? I am not sure if these mutants are included in the Nebraska library though.
3. Sbi is a LTA-binding *S. aureus* protein. Was its release unaffected by MprF and YpfP?

Some minor points:

4. Page 4, second para: It is stated that LukAB are only detected associated with the bacterial cells. However, later it becomes clear that a certain proportion is also secreted.
5. Page 6, third para: Explain Hmmer and ScanProsite.
6. Fig. 1B: Please explain LukS* and explain how LukE, HlgA, and HlgC were detected. Is the antibody cross-reacting with all these proteins?
7. Title: I would replace 'secretion' with 'release' or 'retention' considering that secretion is mostly regarded as the process of translocation across membranes.

We thank the Editor and the Reviewers for their thoughtful evaluation of our manuscript and are pleased to see the overall enthusiasm about our study. After careful consideration of all the comments, we have included a number of new experiments and have edited the text to address each of the concerns (changes are indicated in red in the marked version of the main text). The new experiments, in response to the critiques, have resulted in a much-improved manuscript. Below are point-by-point responses to each of the Reviewers' comments.

REVIEWER COMMENTS:

REVIEWER #1

In the report, the cell envelope of *Staphylococcus aureus* selectively controls the secretion of virulence factors, Zheng and colleagues show that LukAB toxins localize in a reservoir in the cell envelope of *S. aureus* cells. They provide genetic evidence of the mechanism involved in LukAB deployment in the cell envelope. They show other secreted proteins to follow a similar path, arguing that this may represent a multistep secretion system that allow bacteria to generate an exoprotein storage in bacterial cell wall to allow fast response during infections. This is a potentially interesting manuscript. I have comments to specific experiments and controls.

Response: We would like to thank the reviewer for his/her generous comments.

Comment 1. Fig. 1D. Detection of LukA by immunoblotting. Detection of LukB is missing. In panel 1C, the authors detected the two toxins LukA and LukB. Is there a reason why Fig. 1D shows only LukA?

Response: Since LukA and LukB can independently bind to the bacterial cell and be secreted into the culture supernatant (Fig. 1c), we focused on detecting LukA for the rest of the study. We have now added the LukB data to Fig. 1d, and the controls for this experiment are shown in Supplementary Fig. 2e.

Comment 2. The section in page 5 needs a revision. I read it several times and I am still not sure I understood Fig. 2B. In this section, the authors investigated the role of bacterial-associated LukAB by inhibiting bacterial protein synthesis using antibiotics and cocultured with PMN. They showed that, infecting PMN with antibiotics reduced the cytotoxicity of *S. aureus*. It is difficult to see the connection of this result to LukAB expression. I think the authors tried to show that lukAB expression is induced during PMN infection. Antibiotics inhibit LukAB translation and prevented the LukAB production during infection, causing a reduction in *S. aureus* cytotoxicity. I still don't understand the point of this experiment.

Response: We have rephrased the text to make this section clearer. In this experiment, we examined the PMN killing capacity of pre-synthesized bacteria-associated LukAB. To exclude potential effects caused by active LukAB synthesis and production, antibiotics were used to stop protein translation. The LukAB-dependent PMN killing in the presence of antibiotics demonstrate that bacteria-associated LukAB is active and can kill PMNs.

Comment 3. To decouple gene expression and protein production, the authors engineered an inducible strain and showed that, upon artificial lukAB induction, *S. aureus* killed PMN with or

without antibiotics. Here, I think the authors tried to demonstrate that the presence of antibiotic in the killing medium only inhibits the production of new LukAB proteins; the old LukAB protein already attached to the cell wall proved enough to kill PMN. As I said, I'm not sure I understood. Moreover, additional controls are needed to show that LukAB contributes to the killing effect. For instance, mutants in the other leukocidins, which do not accumulate in the cell wall, should not produce the PMN killing effect detected in Fig 2B-D.

Response: The reviewer's interpretation is correct. This experiment is an extension of Fig. 2b by controlling LukAB production tightly by exogenous hemin. Both Fig. 2b and 2c indicate that bacteria-associated LukAB can be deployed to kill PMNs. As the $\Delta lukAB$ mutant we used in Fig. 2b is competent to produce other leukocidins, our results already show that the other leukocidins, which do not accumulate in the cell envelope, are not responsible for PMN death in this model. Nevertheless, we took the reviewer's suggestion to compare LukAB to another leukocidin directly under the same promoter. To this end, we engineered a hemin-inducible LukAB and PVL in a leukocidin-null USA300 isogenic mutant and performed PMN infections in the presence of tetracycline. The new Fig. 2c shows that only LukAB pre-induction can kill PMNs. We have also added Supplementary Fig. 3d-e to show the localization of LukAB and PVL under this condition.

Comment 4. Is it possible to use purified cell wall from WT (which contains LukAB) or maybe purified LukAB to complement $\Delta lukAB$ -PMN interaction? This will probe that LukAB is required for the USA300-PMN and will facilitate to understand this section.

Response: We believe that this comment is addressed with the experiment shown in Fig. 2a. For these experiments, *S. aureus* were treated with lysostaphin to cleave the cell wall and thus lyse the bacterial cells. The resulting soluble fraction, which contains small cell wall fragments and released proteins, was used to intoxicate PMNs. By using a neutralizing anti-LukA antibody as well as the isogenic $\Delta lukAB$ mutant strain as controls, these data confidently establish that cell envelope-associated LukAB is indeed active and responsible for the observed lysis of PMNs.

Comment 5. I have two comments in relation to the fluorescent microscopy experiment presented in page 6. The authors performed immunostaining experiments to localize LukAB signal in *S. aureus* cells. It is well known that *S. aureus* immunostaining produces artifacts as the antibodies bind non-specifically to the staphylococcal cell surface adhesins. To avoid this, the authors performed the experiments in a *Dspa Dsbi* mutant, which lacks two of the most relevant adhesins. The authors show localization of LukAB in large, round foci, which is also the classical signal distribution pattern of non-specific binding. The problem here is that, whether the antibodies bind to LukAB or to remaining cell-exposed adhesins, they may generate the same signal distribution pattern and it would be difficult to determine if it is a legitimate signal.

Two suggestions below to help authors to overcome this issue and to complement their fluorescence microscopy experiments.

The authors should add a supplemental figure in which the signal of the control strain (*Dspa Dsbi*) is shown at different exposure times and compared to that of the sample strain. In the current stage, the Alexa Fluor 594 channel produces a black field. It would be interesting to show images of this channel at different exposure times, to reach the exposure level in which the background signal allows the visualization of the cell shape and still no foci can be distinguished.

The signal should be different in the sample strain, in which different exposure times should show cell wall foci in all cases.

Another option is to obtain a similar signal distribution pattern using another approach that does not involve antibodies. This would validate the signal obtained using antibodies. I recommend the use of SNAP or HALO tags to LukAB. These tags bind fluorophores very specifically and the signal can be traced using the Alexa fluor 594 channel. No antibodies involved. A supplemental figure showing that immunostaining and SNAP staining produce similar signal distribution would be sufficient to validate Figure 3.

Response: First, we would like to clarify that all the strains used for LukA immunostaining are lacking *spa* and *sbi*. In each experiment, signals in the control strain ($\Delta lukAB \Delta spa \Delta sbi$) were set as background and we only considered signal higher than background to be antibody specific signal. We noticed that this background signal is much lower than the LukAB-specific signal. We also would like to point out that in Supplementary Fig. 4c, protein A staining using the same method shows continuous signal around the cell, which is a different pattern from the LukAB foci.

As per the reviewer suggestion, we engineered new SNAP- or HALO- tagged LukAB producing strains to further validate the antibody staining. We were able to visualize surface-exposed SNAP-tagged LukAB with a similar distribution pattern as immunostaining (Response Fig. 1a) despite the fact that this large fusion protein was prone to degradation (only ~50% of the fused protein seems to be intact) (Response Fig. 1b). Of note, the production of HALO-tagged LukAB was too low to be detected (Response Fig. 1b).

Response Figure 1.

a. Imaging of SNAP tagged LukAB at early stationary phase (5h). A SNAP-tag was fused to the N-terminus of LukA and the SNAP-tagged LukAB was expressed under the control of *PhrtAB* promoter in the presence of 2 μ M hemin. The cells were stained with SNAP-surface 594 (NEB S9134S) for the SNAP-tag and BODIPY FL vancomycin for the cell wall. The middle and bottom panels show two representative SNAP-tagged LukAB images. Yellow arrows point to single cells shown in enhanced image on the right. Scale bar, 1 μ m.

b. Immunoblot of native LukA, SNAP-tagged LukA, and HALO-tagged LukA using an anti-LukA polyclonal antibody.

Comment 6. My second comment is related to the use of protoplasts. How can the authors be sure that cells were protoplast, that no cell wall remains in the cells? It is possible to follow the formation of protoplast in rod-shaped bacteria with the change of shape to spheres but in the case of the already spherical *S. aureus*, how could you know that no cell wall remains in the cells?

Response: We have examined the presence of cell wall in these protoplasts and added Supplementary Fig. 4e to show the results. In this experiment, we used an *S. aureus* strain expressing mCherry in the cytoplasm, and stained the cell membrane with FM 4-64 and the cell wall with BODIPY FL vancomycin. When the cell wall was cleaved by lysostaphin, the majority of cell wall was removed and the remainder was present as fragments. The cell membrane of most cells remained intact and the cytoplasmic mCherry remained in the cytoplasm.

Comment 7. The authors performed a genetic screen and identified the *mprF* and *ypfP* genes involved in LukAB cell wall deposition and secretion. These experiments are very interesting. I just have one suggestion to the authors. MprF adds lisil residues to PG. *ypfP* contributes to the LTA formation, which bind alanyl residues and Mg²⁺ cations. Therefore, both *mprF* and *ypfP* genes contribute a net positive charge to the *S. aureus* surface envelope by lysinylation of PG or alanylation of teichoic acids. Possibly, LukAB (as well as the other proteins) is retained in the cell wall as a consequence of the net positive charges that LPG or LTA add to the cell wall. If this is the case, adding Mg²⁺ to the culture medium will add extra net positive charge to the cell wall thus it will complement DmprF and DypfP mutants in their capacity to retain LukAB in the cell wall.

Response: We thank the reviewer for the excellent suggestion. We have now tested the effect of Mg²⁺ addition on the secretion of LukAB. Adding Mg²⁺ to the culture medium reduced the surface positive charge (Response Figure 2a), potentially due to the transcriptional effect on the *Dlt* operon (Koprivnjak et al., 2006). While the Mg²⁺ addition indeed reduced the levels of LukAB in the culture supernatant (Response Figure 2b), this effect seems to be due to decreased *lukAB* transcription (Response Figure 2c). Thus, unfortunately, we can't really conclude anything on whether Mg²⁺ addition directly influences LukAB sorting.

REVIEWER #2

In this study, the authors address how secretion of the leukocidin LukAB, which is known to be both secreted and cell envelope-associated, is controlled by *S. aureus*. *S. aureus* secretes numerous factors important both in its physiology and in virulence. Although how proteins are secreted through the membrane is generally known, there are major gaps in our understanding of how proteins move from the cell membrane to other sites in the cell envelope or beyond. The authors are addressing a fundamental question, the experiments in the manuscript are well done, and I believe that this manuscript will find a receptive audience in those interested in this problem no matter where it is published. However, I do not know whether this manuscript meets the bar for publication in Nature Communications because it still does not provide a mechanism-based understanding. I will leave that up to the editor because I wouldn't mind seeing it published there.

Response: We would like to thank the reviewer for his/her supportive comments.

The key findings of this study are the following: 1) Cell-envelope-associated LukAB is an active toxin. 2) LukAB forms a punctate foci pattern at the cell envelope. 3) LukAB is found in two distinct cell envelope "depots" (surface-exposed and membrane-proximal). 4) LukAB secretion and localization are altered in deletion mutants of two genes, *mprF* and *ypfP*, that encode cell envelope biogenesis enzymes. 5) Other proteins known to be secreted by *S. aureus* are also found in two distinct "depots" and are affected by deletion of *mprF* and *ypfP*.

Major Comments:

Comment 1. The authors frequently discuss the idea of a "multistep process" for LukAB secretion in which LukAB, after translocation across the membrane, is first stored in a membrane-proximal compartment and then a surface-exposed compartment before finally becoming fully dissociated from the cell. While the evidence does clearly suggest that there are two discrete cell envelope-associated "depots" where LukAB is found, evidence for a "multistep process" is lacking. For example, it is unclear whether individual LukAB dimers traverse from the membrane-proximal compartment to the surface-exposed compartment (rather than from the

membrane-proximal compartment straight to the supernatant). Similarly, whether or not all secreted LukAB dimers initially spend any significant amount of time in either compartment is unknown.

Response: While we can't measure the exact time for which LukAB stays in each compartment, the fact that we can detect LukAB, but not other leukocidins, in these cell envelope-associated compartments suggests that LukAB spends more time in either compartment compared to other exoproteins. Regarding the order of this multistep process, we have now included new data (Fig. 4f) to show that LukAB is first located in the membrane-proximal compartment and then translocated to the cell surface and extracellular milieu. It's unclear yet to us whether LukAB is sorted to the surface-exposed compartment before or after the extracellular milieu. However, our results suggest that the process of LukAB trafficking between surface-exposed compartment and extracellular milieu is controlled by YpfP-LtaA.

Comment 2. In the results from the targeted mutant screen, it is clear that deletions of *mprF* and *ypfP* cause defective secretion of LukAB. While *mprF* was found as a hit in the Nebraska library screen, *ypfP* was not. Is there a reason that the *ypfP* mutant did not come up in this screen?

Response: To get to the bottom of this question, we pulled out the *ypfP* mutant from the arrayed transposon library stored in our lab. By performing a PCR on the *ypfP* gene, we found that the strain in the well of the 96-well plate did not contain a transposon insertion in the *ypfP* gene explaining the discrepancy. In contrast, the *ypfP* mutant used in the targeted screen was obtained as an individual validated strain directly from BEIresources.

Comment 3. Additionally, the *ltaA* mutant was observed as a hit for increased LukAB secretion in the Nebraska library screen, which is confusing because LtaA is the flippase that flips the LTA glycolipid to the outer leaflet of the membrane after it is generated by YpfP on the inner leaflet of the membrane. For this reason, YpfP-LtaA should not be referred to as a single entity. Is there a way to reconcile opposing effects of *ypfP* and *ltaA* deletion on LukAB secretion given that they play roles in adjacent steps in LTA biogenesis?

Response: We would like to clarify that in the Nebraska library screen, the *ltaA* mutant exhibits increased LukAB levels in the surface-exposed compartment, not LukAB secretion. Thus, the *ltaA* phenotype is in line with the phenotype seen in the *ypfP* mutant.

Comment 4. Do the authors call it YpfP-LtaA because they can only complement with the full operon? (The *ypfP* deletion may have polar effects on *ltaA*.) They should clarify, and they need to deal with the *ltaA* issue in some manner.

Response: Yes, we have tried complementation studies with *ypfP* alone but only the *ypfP-ltaA* operon fully complemented the phenotype of the *ypfP* mutant. We included this clarification in line 274 in the manuscript.

Comment 5. Also related to the two screens, the authors should clarify in the main text the process they went through to narrow down the initial hits. The Methods section titled "Dot blot screen on the Nebraska library" has a very informative paragraph, and it would be helpful if this information was present in the main text. There are still some aspects that are not clear however. For example, what are the hits found in Figure S6A? There are more than 21 in each set of hits

(for higher and lower signal), though in the Methods section, it says that “21 mutants with most and least LukA signal (total 42 mutants)” were examined? Are these Figure S6A hits a subset of the “161 mutants with LukA signal with higher than 150% and 121 mutants with lower than 50% of the WT signal”? If so, why are these displayed but not the others?

Response: We have now changed the Supplementary Fig. 6a to a diagram to clarify the screening approaches. The information in the original Supplementary Fig. 6a as well as the data from each validation step can be found in the Supplementary Table 1.

To answer this reviewer’s questions, the “161 mutants with LukA signal with higher than 150% and 121 mutants with lower than 50% of the WT signal” were selected based on the primary screen. These mutants were examined by dot blots in three independent experiments as the secondary screen. From the secondary screen, 38 and 28 mutants exhibit consistently higher and lower LukAB signals and were selected. From there, we excluded known transcriptional regulators as well as mutants with strong growth defect and examined “21 mutants with higher and lower LukA signal (total 42 mutants)” for the production of LukAB and other leukocidins by western blots.

Comment 6. There should also be information about the targeted screen - why were these genes chosen, and why was *esaA* looked at in the LAC background? In general, for mutants discarded because they are “known” or have “potential” to regulate *lukAB* transcription, what evidence is there that *lukAB* transcription is affected for these mutants (for the ones with “potential” to regulate, specifically)? It would be beneficial to have a table showing results from all the mutants and why they were or were not discarded for future analysis.

Response: The Supplementary Table 1 has a tab designated for the information regarding the targeted screen. Please see below for the answers to individual question:

A. Why were these genes chosen?

Response: They are all the genes we could think of that are involved in protein transportation, surface structure decoration, or post-secretion modification and that are available from BEI resources (individually validated strain from the transposon library).

B. Why was *esaA* looked at in the LAC background?

Response: The initial screen showed reduced LukAB secretion in the *esaA* mutant in the LAC derivative strain JE2. However, we were unable to reproduce the phenotype after transducing the mutation into a clean LAC background. Thus, we did not perform additional follow-up studies.

C. What evidence is there that *lukAB* transcription is affected for these mutants (for the ones with “potential” to regulate, specifically)?

Response: We consider a mutant to potentially regulate *lukAB* transcription because it shows the following phenotypes: 1) it changed the levels of other toxins in the supernatant in the same way as LukAB, because we only observed this multi-step secretion in LukAB and all the leukocidins are generally transcriptionally co-regulated; 2) the levels of LukAB in the supernatant and associated with the bacterial cell are changed in the same trend. The total production of LukAB was influenced in these cases, and thus we didn’t follow up on those mutants.

Minor Comments/Questions:

Comment 10. General Comments/Questions

A. The term “bacteria-associated” should be replaced with “cell envelope-associated” to be more descriptive.

Response: We have changed “bacteria-associated” to “cell envelope-associated” in text after we show that LukAB is associated with the cell envelope (Fig. 1d).

B. The title should better reflect the specific findings of the paper (something about sorting into two distinct cell envelope compartments, perhaps).

Response: We have changed the title to “The cell envelope of *Staphylococcus aureus* selectively controls the sorting of virulence factors”.

Comment 11. Abstract and Introduction:

A. In the Abstract, “anchored to” in the sentence beginning “Intriguingly, one of the leukocidins...” should be changed to “associated with” or something similar, as “anchored” suggests a covalent attachment.

Response: We have changed “anchored to” to “associated with” in the abstract.

B. In the Abstract the phrase “ready to be deployed” should be removed, as the manuscript does not provide evidence of a “deployment” mechanism or trigger; it could just be that cell envelope-associated LukAB is gradually shed by the cell rather than being specifically “deployed.”

Response: We have removed “ready to be deployed” and changed this sentence to “... retention of LukAB in the cell envelope provides *S. aureus* with a pre-synthesized active toxin that kills immune cells.” in the abstract.

C. Remove “to date” in the first paragraph; this is redundant with “ongoing discussion.”

Response: We have removed “to date” in line 40.

D. Clarify what is known about the component(s) of the cell envelope that GW repeats and LysM domains interact with.

Response: The GW repeats interact with lipoteichoic acid (Baba and Schneewind, 1998; Zoll et al., 2012), and the LysM domain is associated with the repeating disaccharide β -*N*-acetylmuramic acid-(1 → 4)- β -*N*-acetylglucosamine of staphylococcal peptidoglycan (Frankel and Schneewind, 2012).

E. The introduction ends with the statement that the results here “establish the role of the *S. aureus* cell envelope for the sorting and secretion of selective exoproteins.” Other papers have commented on involvement of cell envelope factors such as WTA in “spatial regulation” of secreted proteins such as Atl (DOI: 10.1128/AAC.00323-18; and WTA has also been implicated in spatial regulation of membrane-anchored proteins by an unknown mechanism; the authors’ own data also supports the previous evidence that WTA is somehow involved in sorting/secretion). In addition, the membrane protein LyrA (now called SpdC) was also

implicated in protein secretion (and so were other “surface protein display” factors). The glucosaminidase SagB, which affects PG strand length, was independently implicated in protein secretion. (A very recent Nat Micro paper shows that LyrA/SpdC regulates the activity of SagB, so the two are connected.) Therefore, it was known previously that the *S. aureus* cell envelope is important for sorting and secretion of exoproteins, and the authors do in part acknowledge this elsewhere. So while this manuscript is an interesting, and I think useful, contribution to the growing literature, the last sentence of the introduction overstates the findings. It would be good to rephrase for clarity and perhaps cite previous literature.

Response: We have changed the word “establish” to “highlight” in line 76. We appreciate the citations this reviewer brought up and have included some in lines 397-398 in the discussion. We believe that our study does directly establish the role of cell envelope on the sorting process *en route* of protein secretion by 1) excluding the interference of transcriptional effect and 2) identifying the protein depot when the secretion is hampered.

Comment 12. Comments on the section titled “LukAB is the only leukocidin associated with the bacterial cell.”

A. The title should be changed, as this was already known; the novel findings are the temporal data, the finding that the abnormal localization of LukAB is not due to heterodimer formation, the subcellular localization of LukAB, and the finding that bacterial-associated LukAB is observed *in vivo*.

Response: We have changed the title to “LukAB is associated with the bacterial cell envelope”.

B. The fact that LukS* recognizes the S subunits of the other leukocidins but not that of LukAB suggests (same with LukF* for the F subunits) that there are important differences between LukAB and these other leukocidins at the biochemical level. A comment on notable differences is warranted (if known). It would also be nice to clarify in the figure captions that the other leukocidins run at the same MW as each other, explaining why there’s only one apparent band in the LukS* and LukF* blots.

Response: The amino acid sequence identity of LukED, PVL, HlgAB and HlgCB ranges from 60-80%, while LukAB exhibits only 30-40% identity to the other leukocidins. We have examined the most notable differences between LukAB and other leukocidins (the extra sequence at the N- and C-terminus (DuMont et al., 2014)), but these didn’t influence the secretion of LukAB. We have clarified in the figure legends that the other leukocidins are highly similar in length and protein sequence.

C. The following is definitely not necessary to address but would be interesting if there is remaining sample that can be analyzed without doing another mouse experiment: the other leukocidins are shown to not be bacterially associated *in vitro* - is this true *in vivo* as well?

Response: Unfortunately, we had to use all the samples for the data shown.

Comment 13. Comments on the section titled “Bacteria-associated LukAB contributes to the cytotoxicity of USA300.”

A. Is it known whether the LukAB dimer is required for cytotoxicity, or does each individual protein have some effect on its own?

Response: When we described the identification of LukAB (DuMont et al., 2011), we established that the cytotoxic activity of LukAB requires the heterodimeric LukA and LukB complex.

B. For Figure 2A, is Anti-LukA negative sera available as a control to ensure that other sera components are not responsible for decreasing cytotoxicity?

Response: We have clarified the text to indicate that the anti-LukA used in this experiment is polyclonal anti-LukA antibody affinity purified from rabbit sera. No other sera components are present in this experiment. The text has been changed in line 182.

C. For Figures 2B and 2C, are there non-protein synthesis inhibitors that can be used to differentiate between the effects of just stopping protein synthesis and that of cell growth inhibition? Also, the amount of bacteria added in these assays after inhibitor treatment should be normalized by OD or cfus instead of by volume to control for growth inhibition during antibiotic incubation.

Response: The purpose of this experiment was to show the cytotoxicity of cell envelope-associated LukAB. We chose protein synthesis inhibitors because they are directly linked to protein synthesis and can be used to uncouple the effect of new LukAB synthesis and pre-synthesized LukAB. The use of non-protein synthesis inhibitors is a great idea and something we will consider for future studies.

We apologize for any confusion, but for all these studies, the bacteria were normalized by OD600 before any treatment. We have plated bacteria before and after antibiotic incubation, and no difference in CFU was found.

Comment 14. Comments on the section titled “LukAB forms discrete foci on USA300 cells.”

A. It would be nice to see a control in the experiment for Figure 3A where localization is examined with the anti-LukB antibody to ensure that the same pattern is observed. It would also be interesting to know if these foci still form if only LukA or only LukB is present.

Response: Unfortunately, our anti-LukB antibody is not specific enough to be used in immunofluorescence experiments. The *lukA* and *lukB* are always co-present in the genome and the toxin is only functional when both subunits are co-produced, thus why we only imaged the strains producing the toxin heterodimer.

B. It would be nice to show data analogous to that in Figure 3B but for exponential phase cells rather than late stationary phase cells, and exponential phase cells have more LukAB foci according to the data in Figure 3F.

Response: We have changed Fig. 3a to show the immunofluorescence staining with cells at the early stationary phase (5h), and showed the images with cells from exponential and late stationary phase in the Supplementary Fig. 4a. Interestingly, the foci number was higher in the late stationary phase than the exponential phase, suggesting that immunostaining on the intact cells only reveals LukAB distribution exposed on the bacterial surface.

C. The captions for the histogram figures should state how many cells were analyzed per condition.

Response: We have included the number of cells analyzed in the histogram figure in the figure legends.

D. What phase were the cells in Figure 3C grown to? If they were grown to stationary phase, it would be better to do this analysis in exponential phase as there would be more septum-containing cells. For the current data, it does not seem that there are enough cells (68) to establish a trend.

Response: We replotted the Fig. 3c combining 196 foci present on cells that contain septum. We combined cells from exponential, early stationary, and late stationary phases.

E. It should be clarified for Figure 3H that the FLAG tag is on LukA.

Response: It is now clarified in line 196.

F. For the sentence beginning “To minimize unspecific binding of the antibodies...” it would be good to explain for general readership that *spa* and *sbi* encode two of the most highly abundant proteins on the *S. aureus* cell surface.

Response: We have added a short description of *spa* and *sbi* in lines 167-168.

G. It wasn't clear to me that Fig 3b added much that justifies inclusion in the main text.

Response: We agree with this comment. We have removed this quantification as Fig. 3c is a better comparison of foci number in different growth phases.

Comment 15. Comments on the section titled “LukAB is secreted through a multistep process.”

A. For the terminology “SDS-resistant compartment” and “SDS-susceptible compartment,” it seems that these phrases should be used in this section only and then, afterwards, only “membrane-proximal” and “surface-exposed” should be used which are more informative/clear.

Response: We agree with this suggestion. We have switched “SDS-resistant” and “SDS-susceptible” to “membrane-proximal” and “surface-exposed” respectively throughout the manuscript.

B. Not necessary but would be interesting: is there a way to look at the impact of the SDS-resistant fraction of LukAB on PMN killing?

Response: To address this, we performed new experiments where we used the trypsin treated bacteria (to remove surface-exposed LukAB) to infect PMNs in the presence of tetracycline (to block newly made toxin). The trypsin treated bacteria retain potent PMN killing activity (Response Fig. 3), suggesting that the membrane-proximal LukAB is active.

Comment 16. Comments on the section titled “MprF and YpfP-LtaA contribute to LukAB secretion.”

A. It would be good to bring up the secondary screen and any validation of the screen in the results section.

Response: We summarized the results of secondary screen and screen validations in the Supplementary Table 1.

B. For Figure S6A, the “common” names of the genes should be listed with the gene IDs. A table may be a better way to show the information in this figure.

Response: In Supplementary Table 1, the common names are listed with the gene IDs.

C. Figures 5H and 5I show that *mprF* and *ypfP* mutants are somewhat more cytotoxic than wild-type cells. How does this fit with previous reports that *mprF* and *ypfP* strains (and other strains defective in glycolipid-anchored LTA) are less virulent than wild-type?

Response: Our assay was a short (1hr) extracellular infection of primary human PMNs. With increased levels of LukAB in the cell envelope of the *mprF* and *ypfP* mutants, these mutants are slightly more cytotoxic compared to WT. However, our model didn’t evaluate the effect of the antimicrobial activity of human PMNs on the mutant *S. aureus* strains as only PMNs were present in this system and we didn’t expect phagocytosis-mediated killing to occur in the absence of opsonins.

D. In Figure S6B, what are the bands between the LukA and Hla bands that appear to be present in some of the samples (*mprF*, *tatC*, *srtA*, *ypfP*, *mecA*)? Are they LukA variants?

Response: We are indeed interested in the size variations of LukA in some of the mutants (eg. *srtA*) for future studies. At this point, we think this could be a LukA variant.

E. In the text, for the sentence starting “Without YpfP or LtaA, *S. aureus*...” it would be good to mention what is meant by “abnormal LTA”. For *ypfP* mutants, LTAs are much longer than normal and have a phosphatidylglycerol anchor rather than a diglucosyl-diacylglycerol anchor, and for *ltaA* mutants, LTAs are intermediate in length (compared to WT and *ypfP*) and have some phosphatidylglycerol anchors and some diglucosyl-diacylglycerol anchors.

Response: We have clarified that LTA is longer in the *ypfP* or *ItaA* mutants in line 260.

F. It would be good to mention that *ypfP* cells are larger than wild-type cells, as this could affect why the *ypfP* mutant seems to have increased exoprotein secretion as shown in Figure 5A.

Response: We have normalized all of our experiments by OD600 and thus we don't think the increased exoprotein secretion in the *ypfP* mutants is due to larger cells. Future studies are needed to uncover why deletion of *ypfP* results in increase secretion and/or expression of exoproteins.

G. Fix grammar: "Mutants known or potential to regulate *lukAB* transcription were excluded." Citations to these mutants should also be provided at the end of the sentence. See also the comment in the "major comments" section.

Response: We removed this sentence. The references for known regulatory effects are listed in the Supplementary Table 1. The results suggesting potential transcriptional effect are also included in the Supplementary Table 1.

H. As a control, it would be nice to analyze inactive *ypfP* and *mprF* point mutants to confirm that it is the products of their enzymatic activities that influence *LukAB* localization. *YpfP/UgtP* has been proposed to have other activities related to protein-protein interactions.

Response: We followed the reviewer advice and performed a series of additional studies to probe for the functionality of *MprF* and *YpfP*. For *MprF*, we have included a new supplementary figure (Supplementary Fig. 7d) showing *LukAB* secretion in the presence of different domains and point mutants. We discovered that the *MprF* synthase and flippase domain alone were not able to restore normal *LukAB* secretion. The enzymatic inactive point mutations in the synthase domain hindered *LukAB* secretion, while the inactive point mutations in the flippase domain retained normal *LukAB* secretion. Our results indicate that the enzymatic activity of *MprF* influences *LukAB* secretion, but also suggests a differential role of the synthase and flippase activities in this process.

For *YpfP*, we have added additional data in Supplementary Fig. 7e showing the *LukAB* secretion profile in the *gtaB* mutant. *GtaB* is upstream of *YpfP* in the biosynthesis of LTA and is part of *Glc₂-DAG* synthesis pathway. Our data show that *GtaB* is also required for the secretion of *LukAB* implicating LTA as the factor involved in *LukAB* trafficking rather than a role of *YpfP* in protein-protein interactions.

I. It would be interesting (although not necessary) to test *LukAB* localization in other strains with altered surface charge like a D-alanylation mutant or a WTA mutant.

Response: We followed the reviewer advice and performed additional studies to examine *LukAB* secretion in a WTA mutant ($\Delta tarO$). Although we noticed reduced *LukAB* levels in the culture supernatant in this mutant (Response Figure 4a), the *lukAB* promoter activity is also greatly reduced in this mutant (Response Figure 4b). Thus, we are unable to conclude whether WTA is directly influencing the secretion process.

Unfortunately, we don't have a D-alanylation mutant.

J. Also, cytochrome C is supposed to be sensitive to positive surface charge, not just surface charge and this should be clarified in the text with a citation to the paper that reported the method originally for *S. aureus* (Peschel 1999 JBC). It is obvious how *mprF* deletion reduces positive surface charge, but it's less obvious how *ypfP* deletion increases positive surface charge. Is it due to increased D-ala? Comment.

Response: We thank the reviewer for the corrections. We have changed “surface charge” to “positive surface charge” and the citation was added in the method in line 756.

We don't have direct evidence explaining why the *ypfP* mutant has increased positive charge on the cell surface. It's possible that longer and increased levels of LTA allow more D-Ala decoration and thus increase positive charge. Alternatively, it's possible that increased cell size reduces surface-to-volume ratio and thus less cytochrome C can be bound to the cell surface.

K. The PMN killing assay isn't particularly convincing to me in terms of the claim that “the *mprF* and *ypfP* mutants exhibited increased LukAB-mediated killing of human PMNs compared to WT USA300”... The effect size seems quite small so even though there's statistical significance, I don't think it's all that important; also for the *ypfP* mutant, *lukAB* promoter activity is elevated, so could it be related to the fact that there's more *lukAB* around?

Response: We agree that the effect size is small, and we have changed it to “a slight increase” in line 312. However, the results show the importance of cell envelope-stored LukAB in killing PMNs in this tissue culture infection model, given that the *mprF* and *ypfP* mutants were considered less virulent in other infection models.

We also included chloramphenicol as the protein synthesis inhibitor in this experiment. Under chloramphenicol conditions, the effect of PMN killing is mediated by LukAB in the cell envelope, eliminating any confounding effect of newly produced toxin.

Comment 17. Comments on the section titled “Multistep secretion of other exoproteins.”

A. For Figure 6B, it would be nice to have a His-tagged control protein that is known to not be present in either the SDS-susceptible or SDS-resistant portions but is present in the supernatant—perhaps a His tagged non-LukAB leukocidin?

Response: We constructed a new strain to produce α -toxin, another secreted toxin. The engineered α -toxin contains a C-terminal His-tagged as the other tested exoproteins. We

demonstrate that His-tagged α -toxin is only found in the culture supernatant (this figure is moved to Supplementary Fig. 9a).

B. For the third panel in Figure 7, there should be longer LTA shown as present because *ypfP* deletion does not cause loss of LTA. When revising this panel's LTA, the two small circles indicating the glucose units should be removed to show loss of the glycolipid anchor. In the first panel, the negative charge inside the cell is really confusing because it looks like a negatively charged something is moving to the outside of the cell and somehow becoming positively charged. It would be better to show positively charged L-PG moving from the inside to the outside or to show two steps to convey synthesis and then translocation. Also, the figure suggests low levels of L-PG on the inside compared with the outside. Is this known? LTA contains both negative and positive charges, with the latter at least partially neutralizing the former. It is not clear what happens to the cell envelope in *ypfP* mutants that leads to increased positive charge, but it would seem necessary to somehow represent that there is a difference. This gets at what they actually learned about mechanism. Possibly the long LTAs retain proteins at/near the surface of the envelope, but it could also be that other things change in the *ypfP* mutant that explain the results.

Response: We thank the reviewer for the comments. We have changed Fig. 7 as per suggestions. Specifically, the arrows and the negative charge symbol around MprF are deleted; more L-PG is shown in the inner leaflet than the outer leaflet based on previous studies (Bayer et al., 2015; Yang et al., 2013); longer LTA polymers and loss of glucose units are shown in the *ypfP* mutant; the cell envelope is stained dimmer to present the reduced hydrophobicity in the *ypfP* mutant, which we think is the major connection between the *ypfP* mutant and LukAB secretion.

C. More on Fig 7: I am also not sure about calling this figure a model when there is no suggested mechanism for how *mprF* and *ypfP-ltaA* cause these differences in sorting/secretion (other than a charge-based mechanism for MprF). Without grappling with some kind of mechanism for the *ypfP* results – and then figuring out how to depict a possible mechanism – the third panel of this figure does not clarify anything even if fixed to show the presence of LTA.

Response: Fig. 7 summarizes our results that MprF and YpfP-LtaA influence the sorting of a selective group of exoproteins (Fig. 5-6). Currently, our model is a charge-based mechanism for MprF and a hydrophobicity-based mechanism for YpfP-LtaA. We have added components showing the charge and hydrophobicity change in Fig. 7. However, we don't want to limit the potential mechanisms to charge and hydrophobicity as other mechanisms may also be involved.

D. For Figure S9E, it seems that deletion of *ypfP* causes increased secretion of *IssA*. Is there evidence to show that this is solely due to a transcriptional effect as the text suggests?

Response: *IsaA* is increased in all three fractions in the *ypfP* mutant, suggesting that the production is increased. We don't know whether this is a sole transcriptional effect but we avoided such proteins when focusing on the sorting mechanisms. We have removed "transcriptional effect" from the text as we didn't measure the transcription activity directly.

Comment 18. Comments on Discussion

Phrases like “rapid release of virulence factors” and “masked from host immune surveillance, while available to be fired rapidly in specific environments” should be avoided as these are not supported by the data in the paper and are just speculative claims. Alternatively, rephrase to clarify speculative from established claims.

Response: We have rephrased the sentences in line 376 and 443 to indicate that this is our model and pure speculation.

Comment 19. Comments on Methods

What protease inhibitor are they using?

Response: Halt™ Protease Inhibitor Cocktail (ThermoFisher, cat. 78438). This is now indicated in the method section in lines 514-515.

REVIEWER #3

Zheng et al describe how modification of cell membrane lipids and of extracellular lipoteichoic (LTA) polymers affect the retention vs. release of the bi-component leucocidin LukAB and other secretory proteins in *S. aureus*. These are new, important findings as they may help to understand Gram-positive bacterial mechanisms to store secreted proteins in the cell envelope. LukAB is one of the critical *S. aureus* virulence factors, whose accumulation at the bacterial surface and targeted release may contribute to pathogenicity. The study is sound and well written.

Response: We would like to thank the reviewer for the supportive comments.

It raises a number of question the authors may consider in a revised manuscript:

Comment 1. The findings raise the question which interactions may retain LukAB in the envelope and which mechanisms will lead to release. SDS can release the proteins but this is an unphysiologically harsh treatment. The crucial role of LPG and LTA suggests a role of electrostatic interactions, which have been reported to be the major forces altered by the MprF and YpfP-dependent reactions. The authors could test if addition of high salt concentrations, in particular of calcium or magnesium ions, which are known to interact with the phosphate groups of LPG and LTA can trigger the release. Alternatively, altered pH values of chaotropic solutes such as urea could be tested. Maybe such experiments could reveal a potential in vivo-related condition that would promote the release of LukAB from the cell envelope.

Response: We agree with this reviewer that SDS is a harsh treatment, so we used SDS as a tool to separate surface-exposed and membrane-proximal compartments. As per the reviewer recommendations, we tested a series of other conditions. We have included new data that show the effects of other treatments like urea, which potently released LukAB, the weaker chaotropic agent LiCl, which had weak effect, and MgCl₂, which behaved similar to LiCl (Supplementary Fig. 5c). Future studies will look into how human PMNs induce the release of LukAB from the bacterial cells.

Comment 2. Do the different retained proteins share a specific net charge or another property?

Response: This is a great question and something we have also been thinking about. Most of the proteins described here have positive net charge except for ScaH (pI 6.0). The pIs of LukA

and LukB are 9.3 and 8.4. The pIs of LukF-PV, LukS-PV, and α -toxin are 9.0, 8.9, and 7.9 even though they are not retained in the cell envelope (Fig. 1a and Supplementary Fig. 2b, 9a).

Comment 3. Inactivation of YpfP has been found to have different consequences in different *S. aureus* strains. It does not eliminate LTA but alters its amount and potentially polymer length. Do the authors have an idea what is altered in their mutant? Did mutations in other genes related to LTA biosynthesis have an impact on LukAB retention (e.g. LTA polymerase LtaS or LTA-modifying enzymes DltABCD)? I am not sure if these mutants are included in the Nebraska library though.

Response: The strains used in this study are all in the LAC background. According to Hesser et al., in *S. aureus* LAC-derived *ypfP* mutants, LTA is not eliminated but has longer polymer length (Hesser et al., 2020). We have clarified this effect in line 260 and the features were drawn in the Fig. 7.

Unfortunately, the *ltaS* and *dlt* mutants are not available in the Nebraska library; the *ltaS* mutant is lethal without compensatory mutations. As indicated in response to Reviewer #2, Comment #16-H, we have now tested a mutant on *gtaB*, another enzyme involved in LTA biosynthesis, which works upstream of YpfP, and found similar results as the *ypfP* mutant. Collectively, these data strengthen our model that LTA is involved in LukAB trafficking.

Comment 4. Sbi is a LTA-binding *S. aureus* protein. Was its release unaffected by MprF and YpfP?

Response: We have tried to detect Sbi with a C-terminal His-tag, but the levels of Sbi were too low in our strain background to compare the localization patterns by immunoblots.

Some minor points:

Comment 5. Page 4, second para: It is stated that LukAB are only detected associated with the bacterial cells. However, later it becomes clear that a certain proportion is also secreted.

Response: We apologize for the confusion. LukAB is only detected on the bacterial cells in late stationary phase cultures, while the toxin is both secreted and associated with the bacterial cells at exponential phase. We have now clarified this in the manuscript in lines 94-97.

Comment 6. Page 6, third para: Explain Hmmer and ScanProsite.

Response: Hmmer and ScanProsite are tools to detect motifs and sequence homologs in proteins (<http://hmmer.org>, <https://prosite.expasy.org/scanprosite/>). We have added citations in line 203.

Comment 7. Fig. 1B: Please explain LukS* and explain how LukE, HlgA, and HlgC were detected. Is the antibody cross-reacting with all these proteins?

Response: Yes. The LukS* antibody recognizes LukE, HlgA, HlgC, and LukS-PV because of high sequence identity between these proteins (60-80%). Importantly, this antibody does not recognize LukA. We have clarified this in the figure legend of Fig. 1.

Comment 8. Title: I would replace 'secretion' with 'release' or 'retention' considering that secretion is mostly regarded as the process of translocation across membranes.

Response: We have changed the title to "The cell envelope of *Staphylococcus aureus* selectively controls the sorting of virulence factors".

References:

Baba, T., and Schneewind, O. (1998). Targeting of muralytic enzymes to the cell division site of Gram-positive bacteria: repeat domains direct autolysin to the equatorial surface ring of *Staphylococcus aureus*. *EMBO J* 17, 4639-4646.

Bayer, A.S., Mishra, N.N., Chen, L., Kreiswirth, B.N., Rubio, A., and Yang, S.J. (2015). Frequency and Distribution of Single-Nucleotide Polymorphisms within *mprF* in Methicillin-Resistant *Staphylococcus aureus* Clinical Isolates and Their Role in Cross-Resistance to Daptomycin and Host Defense Antimicrobial Peptides. *Antimicrob Agents Chemother* 59, 4930-4937.

DuMont, A.L., Nygaard, T.K., Watkins, R.L., Smith, A., Kozhaya, L., Kreiswirth, B.N., Shopsin, B., Unutmaz, D., Voyich, J.M., and Torres, V.J. (2011). Characterization of a new cytotoxin that contributes to *Staphylococcus aureus* pathogenesis. *Mol Microbiol* 79, 814-825.

DuMont, A.L., Yoong, P., Liu, X., Day, C.J., Chumbler, N.M., James, D.B., Alonzo, F., 3rd, Bode, N.J., Lacy, D.B., Jennings, M.P., *et al.* (2014). Identification of a crucial residue required for *Staphylococcus aureus* LukAB cytotoxicity and receptor recognition. *Infection and immunity* 82, 1268-1276.

Frankel, M.B., and Schneewind, O. (2012). Determinants of murein hydrolase targeting to cross-wall of *Staphylococcus aureus* peptidoglycan. *J Biol Chem* 287, 10460-10471.

Hesser, A.R., Matano, L.M., Vickery, C.R., Wood, B.M., Santiago, A.G., Morris, H.G., Do, T., Losick, R., and Walker, S. (2020). The length of lipoteichoic acid polymers controls *Staphylococcus aureus* cell size and envelope integrity. *J Bacteriol*.

Koprivnjak, T., Mlakar, V., Swanson, L., Fournier, B., Peschel, A., and Weiss, J.P. (2006). Cation-induced transcriptional regulation of the *dlt* operon of *Staphylococcus aureus*. *J Bacteriol* 188, 3622-3630.

Yang, S.J., Mishra, N.N., Rubio, A., and Bayer, A.S. (2013). Causal role of single nucleotide polymorphisms within the *mprF* gene of *Staphylococcus aureus* in daptomycin resistance. *Antimicrob Agents Chemother* 57, 5658-5664.

Zoll, S., Schlag, M., Shkumatov, A.V., Rautenberg, M., Svergun, D.I., Gotz, F., and Stehle, T. (2012). Ligand-binding properties and conformational dynamics of autolysin repeat domains in staphylococcal cell wall recognition. *J Bacteriol* 194, 3789-3802.

REVIEWER COMMENTS

Reviewer #1 (Remarks to the Author):

I had problems to read the labels of the response figures that were embedded in the PDF file. The authors answered adequately most of my comments but two issues still stand:

In relation to comment 5 from the previous letter. Immunofluorescence of LukAB is still problematic. I recommended the authors to circumvent this by using antibody-free techniques but they did not produce a satisfactory signal. The use of Ab-based detection technique is not ideal either because it is known to produce an artifactual punctate distributions. Maybe related to this, it is intriguing that *mprF* and *ypfP* mutants, which are affected in the LukAB secretion still show a LukAB punctate distribution similar to WT cells. In addition, Protein A is probably not the best negative control, as it is known to show a punctate distribution pattern as well (DeDent et al., 2007 *JBacteriol* 189, 4473-4484). The authors should comment on how a punctate distribution of LukAB is related to the secretion mechanism that involves *mprF* and *YpfP* (it should be all over the cell wall, as it is based on cell wall charge and composition); why the LukAB punctuate distribution is not affected in these mutants and they should considered control protein different from Protein A.

In relation to Comment 6: Based on the new data in Supplemental Figure 4e, it is likely that protoplasts are still walled cells. 1) *S. aureus* cells show uniform size and shape whereas protoplasts usually show distorted shape and uneven size. See the typical phenotype of *S. aureus* protoplasts in Kawai et al., 2018 *Cell* 172, 1038–1049. 2) The absence of osmotically supportive medium should prevent the preservation of protoplasts. 3) Supplemental figure 4e show an important amount of cell wall signal associated with the cells.

Reviewer #2 (Remarks to the Author):

LukAB Paper: Response to the Response to the Reviewers

Very Minor Grammar/Word-choice edits:

In Line 319, please change “contributed” to “contributes.”

In Line 354, please change “defined” to “characterized.”

Major Comments: Comment 1. The authors frequently discuss the idea of a “multistep process” for LukAB secretion in which LukAB, after translocation across the membrane, is first stored in a membrane-proximal compartment and then a surface-exposed compartment before finally becoming fully dissociated from the cell. While the evidence does clearly suggest that there are two discrete cell envelope-associated “depots” where LukAB is found, evidence for a “multistep process” is lacking. For example, it is unclear whether individual LukAB dimers traverse from the membrane-proximal compartment to the surface-exposed compartment (rather than from the membrane-proximal compartment straight to the supernatant). Similarly, whether or not all secreted LukAB dimers initially spend any significant amount of time in either compartment is unknown.

Response: While we can't measure the exact time for which LukAB stays in each compartment, the fact that we can detect LukAB, but not other leukocidins, in these cell envelope-associated compartments suggests that LukAB spends more time in either compartment compared to other exoproteins. Regarding the order of this multistep process, we have now included new data (Fig. 4f) to show that LukAB is first located in the membrane-proximal compartment and then translocated to the cell surface and extracellular milieu. It's unclear yet to us whether LukAB is sorted to the surface-exposed compartment before or after the extracellular milieu. However, our results suggest that the process of LukAB trafficking between surface-exposed compartment and extracellular milieu is controlled by YpfP-LtaA.

Okay.

Comment 2. In the results from the targeted mutant screen, it is clear that deletions of mprF and ypfP cause defective secretion of LukAB. While mprF was found as a hit in the Nebraska library screen, ypfP was not. Is there a reason that the ypfP mutant did not come up in this screen?

Response: To get to the bottom of this question, we pulled out the ypfP mutant from the arrayed transposon library stored in our lab. By performing a PCR on the ypfP gene, we found that the strain in the well of the 96-well plate did not contain a transposon insertion in the ypfP gene explaining the discrepancy. In contrast, the ypfP mutant used in the targeted screen was obtained as an individual validated strain directly from BEIresources.

Okay.

Comment 3. Additionally, the ltaA mutant was observed as a hit for increased LukAB secretion in the Nebraska library screen, which is confusing because LtaA is the flippase that flips the LTA glycolipid to the outer leaflet of the membrane after it is generated by YpfP on the inner leaflet of

the membrane. For this reason, YpfP-LtaA should not be referred to as a single entity. Is there a way to reconcile opposing effects of ypfP and ltaA deletion on LukAB secretion given that they play roles in adjacent steps in LTA biogenesis?

Response: We would like to clarify that in the Nebraska library screen, the ltaA mutant exhibits increased LukAB levels in the surface-exposed compartment, not LukAB secretion. Thus, the ltaA phenotype is in line with the phenotype seen in the ypfP mutant.

Okay.

Comment 4. Do the authors call it YpfP-LtaA because they can only complement with the full operon? (The ypfP deletion may have polar effects on ltaA.) They should clarify, and they need to deal with the ltaA issue in some manner.

Response: Yes, we have tried complementation studies with ypfP alone but only the ypfP-ltaA operon fully complemented the phenotype of the ypfP mutant. We included this clarification in line 274 in the manuscript.

It would be nice to include a brief explanation/speculation as to why the full operon is necessary (possibly mentioning polar effects) so that future use of the term “YpfP-LtaA” is more clear.

Comment 5. Also related to the two screens, the authors should clarify in the main text the process they went through to narrow down the initial hits. The Methods section titled “Dot blot screen on the Nebraska library” has a very informative paragraph, and it would be helpful if this information was present in the main text. There are still some aspects that are not clear however. For example, what are the hits found in Figure S6A? There are more than 21 in each set of hits 7 (for higher and lower signal), though in the Methods section, it says that “21 mutants with most and least LukA signal (total 42 mutants)” were examined? Are these Figure S6A hits a subset of the “161 mutants with LukA signal with higher than 150% and 121 mutants with lower than 50% of the WT signal”? If so, why are these displayed but not the others?

Response: We have now changed the Supplementary Fig. 6a to a diagram to clarify the screening approaches. The information in the original Supplementary Fig. 6a as well as the data from each validation step can be found in the Supplementary Table 1. To answer this reviewer's questions, the “161 mutants with LukA signal with higher than 150% and 121 mutants with lower than 50% of the WT signal” were selected based on the primary screen. These mutants were examined by dot blots in three independent experiments as the secondary screen. From the secondary screen, 38 and 28 mutants exhibit consistently higher and lower LukAB signals and were selected. From there, we excluded known transcriptional regulators as well as mutants with strong growth defect and examined “21 mutants with higher and lower LukA signal (total 42 mutants)” for the production of LukAB and other leukocidins by western blots.

Okay.

Comment 6. There should also be information about the targeted screen - why were these genes chosen, and why was *esaA* looked at in the LAC background? In general, for mutants discarded because they are “known” or have “potential” to regulate *lukAB* transcription, what evidence is there that *lukAB* transcription is affected for these mutants (for the ones with “potential” to regulate, specifically)? It would be beneficial to have a table showing results from all the mutants and why they were or were not discarded for future analysis.

Response: The Supplementary Table 1 has a tab designated for the information regarding the targeted screen.

Okay.

Please see below for the answers to individual question:

A. Why were these genes chosen?

Response: They are all the genes we could think of that are involved in protein transportation, surface structure decoration, or post-secretion modification and that are available from BEIresources (individually validated strain from the transposon library).

Okay.

B. Why was *esaA* looked at in the LAC background?

Response: The initial screen showed reduced *LukAB* secretion in the *esaA* mutant in the LAC derivative strain JE2. However, we were unable to reproduce the phenotype after transducing the mutation into a clean LAC background. Thus, we did not perform additional follow-up studies.

Okay.

C. What evidence is there that *lukAB* transcription is affected for these mutants (for the ones with “potential” to regulate, specifically)?

Response: We consider a mutant to potentially regulate *lukAB* transcription because it shows the following phenotypes: 1) it changed the levels of other toxins in the supernatant in the same way as *LukAB*, because we only observed this multi-step secretion in *LukAB* and all the leukocidins are generally transcriptionally co-regulated; 2) the levels of *LukAB* in the supernatant and associated with the bacterial cell are changed in the same trend. The total production of *LukAB* was influenced in these cases, and thus we didn't follow up on those mutants.

Okay.

Minor Comments/Questions:

Comment 10. General Comments/Questions

A. The term “bacteria-associated” should be replaced with “cell envelope-associated” to be more descriptive.

Response: We have changed “bacteria-associated” to “cell envelope-associated” in text after we show that LukAB is associated with the cell envelope (Fig. 1d).

Okay.

B. The title should better reflect the specific findings of the paper (something about sorting into two distinct cell envelope compartments, perhaps).

Response: We have changed the title to “The cell envelope of *Staphylococcus aureus* selectively controls the sorting of virulence factors”.

Okay.

Comment 11. Abstract and Introduction:

A. In the Abstract, “anchored to” in the sentence beginning “Intriguingly, one of the leukocidins...” should be changed to “associated with” or something similar, as “anchored” suggests a covalent attachment.

Response: We have changed “anchored to” to “associated with” in the abstract.

Okay.

B. In the Abstract the phrase “ready to be deployed” should be removed, as the manuscript does not provide evidence of a “deployment” mechanism or trigger; it could just be that cell envelope-associated LukAB is gradually shed by the cell rather than being specifically “deployed.”

Response: We have removed “ready to be deployed” and changed this sentence to “... retention of LukAB in the cell envelope provides *S. aureus* with a pre-synthesized active toxin that kills immune cells.” in the abstract.

Okay.

C. Remove “to date” in the first paragraph; this is redundant with “ongoing discussion.”

Response: We have removed “to date” in line 40.

Okay.

D. Clarify what is known about the component(s) of the cell envelope that GW repeats and LysM domains interact with.

Response: The GW repeats interact with lipoteichoic acid (Baba and Schneewind, 1998; Zoll et al., 2012), and the LysM domain is associated with the repeating disaccharide β -N-acetylmuramic acid-(1 \rightarrow 4)- β -N-acetylglucosamine of staphylococcal peptidoglycan (Frankel and Schneewind, 2012).

Okay.

E. The introduction ends with the statement that the results here “establish the role of the *S. aureus* cell envelope for the sorting and secretion of selective exoproteins.” Other papers have commented on involvement of cell envelope factors such as WTA in “spatial regulation” of secreted proteins such as Atl (DOI: 10.1128/AAC.00323-18; and WTA has also been implicated in spatial regulation of membrane-anchored proteins by an unknown mechanism; the authors’ own data also supports the previous evidence that WTA is somehow involved in sorting/secretion). In addition, the membrane protein LyrA (now called SpdC) was also 9 implicated in protein secretion (and so were other “surface protein display” factors). The glucosaminidase SagB, which affects PG strand length, was independently implicated in protein secretion. (A very recent Nat Micro paper shows that LyrA/SpdC regulates the activity of SagB, so the two are connected.) Therefore, it was known previously that the *S. aureus* cell envelope is important for sorting and secretion of exoproteins, and the authors do in part acknowledge this elsewhere. So while this manuscript is an interesting, and I think useful, contribution to the growing literature, the last sentence of the introduction overstates the findings. It would be good to rephrase for clarity and perhaps cite previous literature.

Response: We have changed the word “establish” to “highlight” in line 76. We appreciate the citations this reviewer brought up and have included some in lines 397-398 in the discussion. We believe that our study does directly establish the role of cell envelope on the sorting process en route of protein secretion by 1) excluding the interference of transcriptional effect and 2) identifying the protein depot when the secretion is hampered.

Okay.

Comment 12. Comments on the section titled “LukAB is the only leukocidin associated with the bacterial cell.”

A. The title should be changed, as this was already known; the novel findings are the temporal data, the finding that the abnormal localization of LukAB is not due to heterodimer formation, the subcellular localization of LukAB, and the finding that bacterial-associated LukAB is observed in vivo.

Response: We have changed the title to “LukAB is associated with the bacterial cell envelope”.

Okay.

B. The fact that LukS* recognizes the S subunits of the other leukocidins but not that of LukAB suggests (same with LukF* for the F subunits) that there are important differences between LukAB and these other leukocidins at the biochemical level. A comment on notable differences is warranted (if known). It would also be nice to clarify in the figure captions that the other leukocidins run at the same MW as each other, explaining why there's only one apparent band in the LukS* and LukF* blots.

Response: The amino acid sequence identity of LukED, PVL, HlgAB and HlgCB ranges from 60-80%, while LukAB exhibits only 30-40% identity to the other leukocidins. We have examined the most notable differences between LukAB and other leukocidins (the extra sequence at the N- and C-terminus (DuMont et al., 2014)), but these didn't influence the secretion of LukAB. We have clarified in the figure legends that the other leukocidins are highly similar in length and protein sequence.

Okay.

C. The following is definitely not necessary to address but would be interesting if there is remaining sample that can be analyzed without doing another mouse experiment: the other leukocidins are shown to not be bacterially associated in vitro - is this true in vivo as well?

Response: Unfortunately, we had to use all the samples for the data shown.

Okay.

Comment 13. Comments on the section titled "Bacteria-associated LukAB contributes to the cytotoxicity of USA300."

A. Is it known whether the LukAB dimer is required for cytotoxicity, or does each individual protein have some effect on its own? 10

Response: When we described the identification of LukAB (DuMont et al., 2011), we established that the cytotoxic activity of LukAB requires the heterodimeric LukA and LukB complex.

Okay.

B. For Figure 2A, is Anti-LukA negative sera available as a control to ensure that other sera components are not responsible for decreasing cytotoxicity?

Response: We have clarified the text to indicate that the anti-LukA used in this experiment is polyclonal anti-LukA antibody affinity purified from rabbit sera. No other sera components are present in this experiment. The text has been changed in line 182.

Okay.

C. For Figures 2B and 2C, are there non-protein synthesis inhibitors that can be used to differentiate between the effects of just stopping protein synthesis and that of cell growth inhibition? Also, the amount of bacteria added in these assays after inhibitor treatment should be normalized by OD or cfus instead of by volume to control for growth inhibition during antibiotic incubation.

Response: The purpose of this experiment was to show the cytotoxicity of cell envelope associated LukAB. We chose protein synthesis inhibitors because they are directly linked to protein synthesis and can be used to uncouple the effect of new LukAB synthesis and pre synthesized LukAB. The use of non-protein synthesis inhibitors is a great idea and something we will consider for future studies. We apologize for any confusion, but for all these studies, the bacteria were normalized by OD600 before any treatment. We have plated bacteria before and after antibiotic incubation, and no difference in CFU was found.

Okay.

Comment 14. Comments on the section titled "LukAB forms discrete foci on USA300 cells."

A. It would be nice to see a control in the experiment for Figure 3A where localization is examined with the anti-LukB antibody to ensure that the same pattern is observed. It would also be interesting to know if these foci still form if only LukA or only LukB is present.

Response: Unfortunately, our anti-LukB antibody is not specific enough to be used in immunofluorescence experiments. The lukA and lukB are always co-present in the genome and the toxin is only functional when both subunits are co-produced, thus why we only imaged the strains producing the toxin heterodimer.

Okay.

B. It would be nice to show data analogous to that in Figure 3B but for exponential phase cells rather than late stationary phase cells, and exponential phase cells have more LukAB foci according to the data in Figure 3F.

Response: We have changed Fig. 3a to show the immunofluorescence staining with cells at the early stationary phase (5h), and showed the images with cells from exponential and late stationary phase in the Supplementary Fig. 4a. Interestingly, the foci number was higher in the late stationary phase than the exponential phase, suggesting that immunostaining on the intact cells only reveals LukAB distribution exposed on the bacterial surface.

Okay.

C. The captions for the histogram figures should state how many cells were analyzed per condition. 11

Response: We have included the number of cells analyzed in the histogram figure in the figure legends.

Okay.

D. What phase were the cells in Figure 3C grown to? If they were grown to stationary phase, it would be better to do this analysis in exponential phase as there would be more septum-containing cells. For the current data, it does not seem that there are enough cells (68) to establish a trend.

Response: We replotted the Fig. 3c combining 196 foci present on cells that contain septum. We combined cells from exponential, early stationary, and late stationary phases.

Okay.

E. It should be clarified for Figure 3H that the FLAG tag is on LukA.

Response: It is now clarified in line 196.

Okay.

F. For the sentence beginning “To minimize unspecific binding of the antibodies...” it would be good to explain for general readership that spa and sbi encode two of the most highly abundant proteins on the *S. aureus* cell surface.

Response: We have added a short description of spa and sbi in lines 167-168.

Okay.

G. It wasn't clear to me that Fig 3b added much that justifies inclusion in the main text.

Response: We agree with this comment. We have removed this quantification as Fig. 3c is a better comparison of foci number in different growth phases.

Okay.

Comment 15. Comments on the section titled “LukAB is secreted through a multistep process.”

A. For the terminology “SDS-resistant compartment” and “SDS-susceptible compartment,” it seems that these phrases should be used in this section only and then, afterwards, only “membrane-proximal” and “surface-exposed” should be used which are more informative/clear.

Response: We agree with this suggestion. We have switched “SDS-resistant” and “SDSsusceptible” to “membrane-proximal” and “surface-exposed” respectively throughout the manuscript.

Okay.

B. Not necessary but would be interesting: is there a way to look at the impact of the SDSresistant fraction of LukAB on PMN killing?

Response: To address this, we performed new experiments where we used the trypsin treated bacteria (to remove surface-exposed LukAB) to infect PMNs in the presence of tetracycline (to block newly made toxin). The trypsin treated bacteria retain potent PMN killing activity (Response Fig. 3), suggesting that the membrane-proximal LukAB is active.

Okay.

Comment 16. Comments on the section titled “MprF and YpfP-LtaA contribute to LukAB secretion.”

A. It would be good to bring up the secondary screen and any validation of the screen in the results section.

Response: We summarized the results of secondary screen and screen validations in the Supplementary Table 1.

Okay.

B. For Figure S6A, the “common” names of the genes should be listed with the gene IDs. A table may be a better way to show the information in this figure.

Response: In Supplementary Table 1, the common names are listed with the gene IDs.

Okay.

C. Figures 5H and 5I show that mprF and ypfP mutants are somewhat more cytotoxic than wildtype cells. How does this fit with previous reports that mprF and ypfP strains (and other strains defective in glycolipid-anchored LTA) are less virulent than wild-type?

Response: Our assay was a short (1hr) extracellular infection of primary human PMNs. With increased levels of LukAB in the cell envelope of the mprF and ypfP mutants, these mutants are slightly more cytotoxic compared to WT. However, our model didn't evaluate the effect of the antimicrobial activity of human PMNs on the mutant *S. aureus* strains as only PMNs were present in this system and we didn't expect phagocytosis-mediated killing to occur in the absence of opsonins.

Okay.

D. In Figure S6B, what are the bands between the LukA and Hla bands that appear to be present in some of the samples (mprF, tatC, srtA, ypfP, mecA)? Are they LukA variants?

Response: We are indeed interested in the size variations of LukA in some of the mutants (eg. srtA) for future studies. At this point, we think this could be a LukA variant.

Okay.

E. In the text, for the sentence starting “Without YpfP or LtaA, *S. aureus*...” it would be good to mention what is meant by “abnormal LTA”. For ypfP mutants, LTAs are much longer than normal and have a phosphatidylglycerol anchor rather than a diglucosyl-diacylglycerol anchor, and for ItaA mutants, LTAs are intermediate in length (compared to WT and ypfP) and have some phosphatidylglycerol anchors and some diglucosyl-diacylglycerol anchors.

Response: We have clarified that LTA is longer in the ypfP or ItaA mutants in line 260.

Okay.

F. It would be good to mention that ypfP cells are larger than wild-type cells, as this could affect why the ypfP mutant seems to have increased exoprotein secretion as shown in Figure 5A.

Response: We have normalized all of our experiments by OD600 and thus we don't think the increased exoprotein secretion in the ypfP mutants is due to larger cells. Future studies are needed to uncover why deletion of ypfP results in increase secretion and/or expression of exoproteins.

Okay.

G. Fix grammar: “Mutants known or potential to regulate lukAB transcription were excluded.” Citations to these mutants should also be provided at the end of the sentence. See also the comment in the “major comments” section.

Response: We removed this sentence. The references for known regulatory effects are listed in the Supplementary Table 1. The results suggesting potential transcriptional effect are also included in the Supplementary Table 1.

Okay.

H. As a control, it would be nice to analyze inactive ypfP and mprF point mutants to confirm that it is the products of their enzymatic activities that influence LukAB localization. YpfP/UgtP has been proposed to have other activities related to protein-protein interactions.

Response: We followed the reviewer advice and performed a series of additional studies to probe for the functionality of MprF and YpfP. For MprF, we have included a new supplementary figure (Supplementary Fig. 7d) showing LukAB secretion in the presence of different domains and point mutants. We discovered that the MprF synthase and flippase domain alone were not able to restore normal LukAB secretion. The enzymatic inactive point mutations in the synthase domain hindered LukAB secretion, while the inactive point mutations in the flippase domain retained normal LukAB secretion. Our results indicate that the enzymatic activity of MprF influences LukAB secretion, but also suggests a differential role of the synthase and flippase activities in this process. For YpfP, we have added additional data in Supplementary Fig. 7e showing the LukAB secretion profile in the *gtaB* mutant. *GtaB* is upstream of YpfP in the biosynthesis of LTA and is part of Glc2-DAG synthesis pathway. Our data show that *GtaB* is also required for the secretion of LukAB implicating LTA as the factor involved in LukAB trafficking rather than a role of YpfP in protein-protein interactions.

Okay.

I. It would be interesting (although not necessary) to test LukAB localization in other strains with altered surface charge like a D-alanylation mutant or a WTA mutant.

Response: We followed the reviewer advice and performed additional studies to examine LukAB secretion in a WTA mutant ($\Delta tarO$). Although we noticed reduced LukAB levels in the culture supernatant in this mutant (Response Figure 4a), the LukAB promoter activity is also greatly reduced in this mutant (Response Figure 4b). Thus, we are unable to conclude whether WTA is directly influencing the secretion process. Unfortunately, we don't have a D-alanylation mutant.

Okay.

J. Also, cytochrome C is supposed to be sensitive to positive surface charge, not just surface charge and this should be clarified in the text with a citation to the paper that reported the method originally for *S. aureus* (Peschel 1999 JBC). It is obvious how *mprF* deletion reduces positive surface charge, but it's less obvious how *ypfP* deletion increases positive surface charge. Is it due to increased D-ala? Comment.

Response: We thank the reviewer for the corrections. We have changed "surface charge" to "positive surface charge" and the citation was added in the method in line 756. We don't have direct evidence explaining why the *ypfP* mutant has increased positive charge on the cell surface. It's possible that longer and increased levels of LTA allow more D-Ala decoration and thus increase positive charge. Alternatively, it's possible that increased cell size reduces surface-to-volume ratio and thus less cytochrome C can be bound to the cell surface.

Okay.

K. The PMN killing assay isn't particularly convincing to me in terms of the claim that "the *mprF* and *ypfP* mutants exhibited increased LukAB-mediated killing of human PMNs compared to WT USA300"... The effect size seems quite small so even though there's statistical significance, I don't think it's all that important; also for the *ypfP* mutant, *lukAB* promoter activity is elevated, so could it be related to the fact that there's more *lukAB* around?

Response: We agree that the effect size is small, and we have changed it to "a slight increase" in line 312. However, the results show the importance of cell envelope-stored LukAB in killing PMNs in this tissue culture infection model, given that the *mprF* and *ypfP* mutants were considered less virulent in other infection models. We also included chloramphenicol as the protein synthesis inhibitor in this experiment. Under chloramphenicol conditions, the effect of PMN killing is mediated by LukAB in the cell envelope, eliminating any confounding effect of newly produced toxin.

Okay.

Comment 17. Comments on the section titled "Multistep secretion of other exoproteins."

A. For Figure 6B, it would be nice to have a His-tagged control protein that is known to not be present in either the SDS-susceptible or SDS-resistant portions but is present in the supernatant- -perhaps a His tagged non-LukAB leukocidin?

Response: We constructed a new strain to produce α -toxin, another secreted toxin. The engineered α -toxin contains a C-terminal His-tagged as the other tested exoproteins. We demonstrate that His-tagged α -toxin is only found in the culture supernatant (this figure is moved to Supplementary Fig. 9a).

Okay.

B. For the third panel in Figure 7, there should be longer LTA shown as present because *ypfP* deletion does not cause loss of LTA. When revising this panel's LTA, the two small circles indicating the glucose units should be removed to show loss of the glycolipid anchor. In the first panel, the negative charge inside the cell is really confusing because it looks like a negatively charged something is moving to the outside of the cell and somehow becoming positively charged. It would be better to show positively charge L-PG moving from the inside to the outside or to show two steps to convey synthesis and then translocation. Also, the figure suggests low levels of L-PG on the inside compared with the outside. Is this known? LTA contains both negative and positive charges, with the latter at least partially neutralizing the former. It is not clear what happens to the cell envelope in *ypfP* mutants that leads to increased positive charge, but it would seem necessary to somehow represent that there is a difference. This gets at what they actually learned about mechanism. Possibly the long LTAs retain proteins at/near the

surface of the envelope, but it could also be that other things change in the ypfP mutant that explain the results.

Response: We thank the reviewer for the comments. We have changed Fig. 7 as per suggestions. Specifically, the arrows and the negative charge symbol around MprF are deleted; more L-PG is shown in the inner leaflet than the outer leaflet based on previous studies (Bayer et al., 2015; Yang et al., 2013); longer LTA polymers and loss of glucose units are shown in the ypfP mutant; the cell envelope is stained dimmer to present the reduced hydrophobicity in the ypfP mutant, which we think is the major connection between the ypfP mutant and LukAB secretion.

We agree with the changes made, but in the case of the dimmer staining to represent reduced hydrophobicity, the difference is not very apparent, especially when printed, so it may be good to either change the color (and add to the key showing what the color means) or use a different pattern, or increase the contrast in dimness further.

C. More on Fig 7: I am also not sure about calling this figure a model when there is no suggested mechanism for how mprF and ypfP-ltaA cause these differences in sorting/secretion (other than a charge-based mechanism for MprF). Without grappling with some kind of mechanism for the ypfP results – and then figuring out how to depict a possible mechanism – the third panel of this figure does not clarify anything even if fixed to show the presence of LTA.

Response: Fig. 7 summarizes our results that MprF and YpfP-LtaA influence the sorting of a selective group of exoproteins (Fig. 5-6). Currently, our model is a charge-based mechanism for MprF and a hydrophobicity-based mechanism for YpfP-LtaA. We have added components showing the charge and hydrophobicity change in Fig. 7. However, we don't want to limit the potential mechanisms to charge and hydrophobicity as other mechanisms may also be involved.

Okay.

D. For Figure S9E, it seems that deletion of ypfP causes increased secretion of IssA. Is there evidence to show that this is solely due to a transcriptional effect as the text suggests?

Response: IssA is increased in all three fractions in the ypfP mutant, suggesting that the production is increased. We don't know whether this is a sole transcriptional effect but we avoided such proteins when focusing on the sorting mechanisms. We have removed "transcriptional effect" from the text as we didn't measure the transcription activity directly.

Okay.

Comment 18. Comments on Discussion 16 Phrases like "rapid release of virulence factors" and "masked from host immune surveillance, while available to be fired rapidly in specific environments" should be avoided as these are not supported by the data in the paper and are just speculative claims. Alternatively, rephrase to clarify speculative from established claims.

Response: We have rephrased the sentences in line 376 and 443 to indicate that this is our model and pure speculation.

Okay.

Comment 19. Comments on Methods What protease inhibitor are they using?

Response: Halt™ Protease Inhibitor Cocktail (ThermoFisher, cat. 78438). This is now indicated in the method section in lines 514-515.

Okay.

Reviewer #3 (Remarks to the Author):

The authors did a great job in revising the manuscript, all my major concerns were appropriately addressed. I have only two minor points left:

- It would be better to restrict the term 'secretion' to processes of translocation across the cytoplasmic membrane (type I-VIII secretion systems...) throughout the text. In contrast, MprF and YpfP appear to govern a subsequent retention/release mechanism. I appreciate the change in the title and suggest to use the terms in similar ways at other positions in the manuscript.
- It is interesting to note that the proteins that behave similarly to LukAB do not share a particular net charge. It might be worth mentioning this point in the discussion to indicate that more subtle molecular properties seem to shape the retention.

REVIEWER COMMENTS:

Reviewer #1 (Remarks to the Author):

Comment 1. I had problems to read the labels of the response figures that were embedded in the PDF file. The authors answered adequately most of my comments but two issues still stand:

Response: We are sorry that the reviewer was unable to read the labels of the prior response. We have enhanced the labeling and included the SNAP-tagged LukAB images below (Response Fig. 1).

Comment 2. In relation to comment 5 from the previous letter. Immunofluorescence of LukAB is still problematic. I recommended the authors to circumvent this by using antibody-free techniques but they did not produce a satisfactory signal. The use of Ab-based detection

technique is not ideal either because it is known to produce an artifactual punctate distributions. Maybe related to this, it is intriguing that *mprF* and *ypfP* mutants, which are affected in the LukAB secretion still show a LukAB punctate distribution similar to WT cells. In addition, Protein A is probably not the best negative control, as it is known to show a punctate distribution pattern as well (DeDent et al., 2007 J Bacteriol 189, 4473-4484). The authors should comment on how a punctate distribution of LukAB is related to the secretion mechanism that involves *mprF* and *YpfP* (it should be all over the cell wall, as it is based on cell wall charge and composition); why the LukAB punctate distribution is not affected in these mutants and they should be considered control protein different from Protein A.

Response: We respectfully disagree with the statement that “using antibody-free techniques but they did not produce a satisfactory signal”. In the prior version of our response, we showed that using SNAP-tagged LukAB (antibody-free staining) as requested by this reviewer, the localization of LukAB on the cell surface is punctate as we showed using antibodies. To extend these data, we have now repeated the SNAP-tagged LukAB studies at two timepoints. The new data show that at both exponential phase (3h) and early stationary phase (5h), the localization of SNAP-tagged LukAB on the cell surface is punctate. Altogether, regardless if the toxin is detected with different antibodies or antibody-free staining, our data show that LukAB exhibits punctate localization on the bacterial surface. The SNAP-tagged LukAB data is now included as Supplemental Material (Supplemental Fig. 4d, e) and discussed in the text (Lines 178-180).

Regarding the *mprF* and *ypfP* mutants, we have now performed careful quantification of isolated single cells from all our immunofluorescence staining images of the WT, *mprF* mutant, and *ypfP* mutant. While the distribution of LukAB remains punctate in the two mutants, we observed subtle differences between the WT and the mutants. In the *mprF* mutant, increased LukAB signal was observed, represented by both increased foci number per cell and increased

intensity of each focus. In the *ypfP* mutant, the number of LukAB foci increased but the intensity of each focus decreased. The increased number of foci, especially in the *ypfP* mutant, suggests that although these two mutants are not the determinants for the discrete localization of LukAB in the bacterial cell envelope, they can both regulate the LukAB distribution. We updated Supplementary Fig. 7f-h and text (Lines 301-309 and 389) to show and discuss these results.

We would like to clarify the reviewer's statement that protein A is not the best negative control. In the experiments pointed out by the reviewer, the cells were first treated with trypsin to remove all surface-exposed proteins, and then incubated for 10-20min before imaging. After 40-60min, the protein A signal covers most of the surface (Fig. 1B in DeDent et al., 2007, doi: 10.1128/JB.00227-07). We didn't treat our bacteria with trypsin before staining, and observed heterogenous but continuous protein A signal on the cell surface, similar to the results reported before (Fig. 1A, 2A in DeDent et al., 2007, doi: 10.1128/JB.00227-07; Fig. 6d in Yu et al., 2018, doi: 10.7554/eLife.34092). Therefore, protein A indeed works well as a control to show that the punctate phenotype of LukAB is not what we see with at least a well-studied surface protein using the same method.

Comment 3. In relation to Comment 6: Based on the new data in Supplemental Figure 4e, it is likely that protoplasts are still walled cells. 1) *S. aureus* cells show uniform size and shape whereas protoplasts usually show distorted shape and uneven size. See the typical phenotype of *S. aureus* protoplasts in Kawai et al., 2018 Cell 172, 1038–1049. 2) The absence of osmotically supportive medium should prevent the preservation of protoplasts. 3) Supplemental figure 4e show an important amount of cell wall signal associated with the cells.

Response: We have avoided using the word "protoplast" as our goal here is to remove most of the cell wall allowing the antibody to access LukAB inside the cell wall. We think that we had sufficient cell wall removal because:

- 1) Comparing the +/- lysostaphin treated cells in the brightfield images in Supplementary Fig. 4e (as well as comparing Fig. 3a and Fig. 3b), the lysostaphin-treated cells are more translucent than the untreated cells, suggesting structural changes. We didn't expect a huge size and shape change as these cells were fixed before lysostaphin treatment, whereas the L-form bacteria shown in the reference pointed out by the reviewer (Kawai et al., 2018, doi: 10.1016/j.cell.2018.01.021) are alive and growing cells.
- 2) Our lysostaphin-treated bacteria are very sensitive to hypotonic conditions. For the immunostaining with these cells, we carried all the steps with an osmotically supportive medium TSM (10 mM MgCl₂, 500 mM Sucrose in 50 mM Tris, pH 7.5). Our initial trial of immunostaining lysostaphin-treated bacteria in PBS resulted in the lysis of all cells.
- 3) Even in the osmoprotective condition, we observed the lysis of some cells during the immunostaining process (see examples in Response Fig. 2), indicating that the lysostaphin-treated cells are fragile without cell wall protection. Most of the residue cell wall signal was found in the clustered cells (potentially due to accessibility of lysostaphin) or the lysed cells (potentially because the released cytoplasmic materials formed non-covalent interactions with the cell wall fragments). We didn't include these

two types of cells in the downstream analyses. The intact cells which we used for analysis had minimal cell wall signal.

We have clarified these points in the manuscript (Lines 182-186).

Reviewer #2 (Remarks to the Author):

I have attached the response to the response to the reviewers (there are only very minor revisions).

Response: We would like to thank the reviewer for his/her supportive comments.

Comment 1. Very Minor Grammar/Word-choice edits: In Line 319, please change “contributed” to “contributes.” In Line 354, please change “defined” to “characterized.”

Response: The text has been modified accordingly (updated line number 315 and 349).

Comment 2. It would be nice to include a brief explanation/speculation as to why the full operon is necessary (possibly mentioning polar effects) so that future use of the term “YpfP-LtaA” is more clear.

Response: We have now included the possibility of the polar effect in line 275.

Comment 3. We agree with the changes made, but in the case of the dimmer staining to represent reduced hydrophobicity, the difference is not very apparent, especially when printed, so it may be good to either change the color (and add to the key showing what the color means) or use a different pattern, or increase the contrast in dimness further.”

Response: We added a roughened pattern to the cell wall of the *ypfP* mutant to clarify the difference.

Reviewer #3 (Remarks to the Author):

The authors did a great job in revising the manuscript, all my major concerns were appropriately addressed. I have only two minor points left:

Response: We would like to thank the reviewer for his/her supportive comments.

- It would be better to restrict the term 'secretion' to processes of translocation across the cytoplasmic membrane (type I-VIII secretion systems...) throughout the text. In contrast, MprF and YpfP appear to govern a subsequent retention/release mechanism. I appreciate the change in the title and suggest to use the terms in similar ways at other positions in the manuscript.

Response: The text has been modified throughout to clarify the role of MprF and YpfP in protein sorting post membrane translocation.

- It is interesting to note that the proteins that behave similarly to LukAB do not share a particular net charge. It might be worth mentioning this point in the discussion to indicate that more subtle molecular properties seem to shape the retention.

Response: We have now included in the discussion these proteins have diverse net charges. Indeed, other features in addition to protein pI may also contribute to protein retention.

REVIEWERS' COMMENTS

Reviewer #1 (Remarks to the Author):

All the issues have been clarified in this revised version.
I have no more questions.